# Exact Optimality of Communication-Privacy-Utility Tradeoffs in Distributed Mean Estimation

**Berivan Isik**
Stanford University
berivan.isik@stanford.edu

**Wei-Ning Chen**
Stanford University
wnchen@stanford.edu

**Ayfer Ozgur**
Stanford University
aozgur@stanford.edu

**Tsachy Weissman**
Stanford University
tsachy@stanford.edu

**Albert No**
Hongik University
albertno@hongik.ac.kr

## Abstract

We study the mean estimation problem under communication and local differential privacy constraints. While previous work has proposed *order*-optimal algorithms for the same problem (i.e., asymptotically optimal as we spend more bits), *exact* optimality (in the non-asymptotic setting) still has not been achieved. In this work, we take a step towards characterizing the *exact*-optimal approach in the presence of shared randomness (a random variable shared between the server and the user) and identify several conditions for *exact* optimality. We prove that one of the conditions is to utilize a rotationally symmetric shared random codebook. Based on this, we propose a randomization mechanism where the codebook is a randomly rotated simplex – satisfying the properties of the *exact*-optimal codebook. The proposed mechanism is based on a $k$-closest encoding which we prove to be *exact*-optimal for the randomly rotated simplex codebook.

## 1 Introduction

The distributed mean estimation problem has attracted attention from the machine learning community as it is a canonical statistical formulation for many stochastic optimization problems such as distributed SGD [1, 3, 31, 32] and federated learning [33, 34]. As these tasks require data collection from the users, the mean estimation problem has often been studied under privacy constraints to protect users' sensitive information. More specifically, several works [2, 4, 7, 8, 9, 29, 35] have analyzed and improved the tradeoff between the utility and $\varepsilon$-local differential privacy ($\varepsilon$-LDP) – the predominant paradigm in privacy mechanisms, which guarantees that an adversary cannot distinguish the user data from the outcome of the privacy mechanism [10, 24]. Among them, [4, 8, 9] developed algorithms that are asymptotically optimal, achieving an optimal mean squared error (MSE) proportional to $\Theta\left(\frac{d}{n \min(\varepsilon, \varepsilon^2)}\right)$, where $n$ is the number of users, and $d$ is the input dimension. Later, [7] proved the corresponding lower bounds that hold for all privacy regimes. However, only `PrivUnit` [4] enjoys *exact* optimality among a large family of mechanisms, as proved by [2], while others provide only *order* optimality and their performance in practice depends heavily on the constant factors.

Another important consideration in the applications of mean estimation is the communication cost during user data collection. For instance, in federated learning, clients need to send overparameterized machine learning models at every round, which becomes a significant bottleneck due to limited resources and bandwidth available to the clients [22, 25, 28]. This has motivated extensive research on mean estimation [31, 34, 38] and distributed SGD [1, 3, 13, 26, 37] under communication constraints; and communication-efficient federated learning [17, 18, 21, 33].

37th Conference on Neural Information Processing Systems (NeurIPS 2023).

In addition to the lines of work that studied these constraints (either privacy or communication) separately, recently, there has also been advancement in the joint problem of mean estimation under both privacy and communication constraints. [6] introduced an *order*-optimal mechanism SQKR requiring only $O(\varepsilon)$ bits by using shared randomness – a random variable shared between the server and the user (see Section 2 for the formal definition). Later, [30] demonstrated better MSE with another *order*-optimal algorithm, MMRC, by simulating PrivUnit using an importance sampling technique [5, 15] – again with shared randomness. In the absence of shared randomness, the *order*-optimal mechanisms proposed by [6] do not achieve the best-known accuracy under this setting and are outperformed by the lossless compression approach in [12] that compresses PrivUnit using a pseudorandom generator (PRG). Due to not using shared randomness, these mechanisms require more bits than others [6, 30] that use shared randomness in the scenarios where it is actually available.

## 1.1 Contributions

To our knowledge, no existing mechanism achieves *exact* optimality under both privacy and communication constraints with shared randomness[1]. In this work, we address this gap by treating the joint problem as a lossy compression problem under $\varepsilon$-LDP constraints.

Our first contribution is to demonstrate that the *exact* optimal scheme with shared randomness can be represented as random coding with a codebook-generating distribution. Specifically, under $b$ bits of communication constraint, the server and the user generate a codebook consisting of $M = 2^b$ vectors (codewords) using shared randomness. The user then selects an index of a vector under a distribution that satisfies $\varepsilon$-LDP constraints, and the server claims the corresponding vector upon receiving the index. We term this approach as "random coding with a codebook" and demonstrate that this (random codebook generation) is the optimal way to use shared randomness.

Next, we prove that the *exact*-optimal codebook-generating distribution must be rotationally symmetric. In other words, for any codebook-generating distribution, the distribution remains the same after random rotation. Based on this insight, we propose Random Rotating Simplex Coding (RRSC), where the codebook-generating distribution is a uniformly randomly rotating simplex. This choice of codebook distribution is reasonable as it maximizes the separation between codewords, which efficiently covers the sphere. The corresponding encoding scheme is the $k$-closest encoding, where the top-$k$ closest codewords to the input obtain high sampling probability, and the remaining ones are assigned low probabilities. We show that this scheme is *exact*-optimal for the random rotating simplex codebook.

The proposed codebook generation is valid only when $M < d$ (or $b \leq \log d$ where $b$ is the communication budget) due to the simplex structure of the codebook. Note that as shown in [6], $b \leq \log d$ bits of communication budget is sufficient to achieve orderwise optimal MSEs under an $\varepsilon$-LDP constraint for any $\varepsilon \leq O(\log d)$, which is usually a common scenario in practical applications such as federated learning where $d$ can range from millions to billions. In addition, we can also extend the scheme for cases when $M \geq d$, by using a codebook consisting of (nearly) maximally separated $M$ vectors on the sphere. As the number of bits $b$ used for communication increases, we demonstrate that the proposed scheme approaches PrivUnit, which is the *exact*-optimal scheme without communication constraints.

Finally, through empirical comparisons, we demonstrate that RRSC outperforms the existing *order*-optimal methods such as SQKR [6] and MMRC [30]. We also observe that the performance of RRSC is remarkably close to that of PrivUnit when the number of bits is set to $b = \epsilon$.

## 1.2 Related Work

The $\ell_2$ mean estimation problem is a canonical statistical formulation for many distributed stochastic optimization methods, such as communication (memory)-efficient SGD [31, 34] or private SGD [24]. For instance, as shown in [14], as long as the final estimator of the mean is unbiased, the $\ell_2$ estimation error (i.e., the variance) determines the convergence rate of the distributed SGD. As a result, there is a long thread of works that study the mean estimation problem under communication constraints [3, 8, 13, 31, 34, 39], privacy constraints [2, 16], or a joint of both [1, 6, 12, 30].

---

[1]Note that we can also eliminate shared randomness with a private coin setting. See Section 5 for a discussion.

Among them, [6] shows that $\Theta(\varepsilon)$ bits are sufficient to achieve the *order*-optimal MSE $\Theta\left(\frac{d}{n\min(\varepsilon,\varepsilon^2)}\right)$ and proposes `SQKR`, an *order*-optimal mean estimation scheme under both privacy and communication constraints. Notice that the MSE of `SQKR` is *orderwise* optimal up to a constant factor. Later on, in [12], it is shown that the pre-constant factor in `SQKR` is indeed suboptimal, resulting in an unignorable gap in the MSE compared to `PrivUnit` – an optimal $\ell_2$ mean estimation scheme under $\varepsilon$-LDP. In the original `PrivUnit`, the output space is a $d$-dimensional sphere $\mathbb{S}^{d-1}$ and hence requires $O(d)$ bits of communication, which is far from the optimal $O(\varepsilon)$ communication bound. However, [12] shows that one can (almost) losslessly compress `PrivUnit` via a pseudo-random generator (PRG). Under the assumption of an existing exponentially strong PRG, [12] proves that one can compress the output of `PrivUnit` using polylog$(d)$ bits with negligible performance loss. Similarly, [30] shows that with the help of shared randomness, `PrivUnit` can be (nearly) losslessly compressed to $\Theta(\varepsilon)$ bits via a channel simulation technique, called `MMRC`. We remark that although the privacy-utility trade-offs in [12] and [30] are (nearly) *exactly* optimal, the communication efficiency is only *order*-optimal. That is, under an exact $b$-bit communication constraint, the MSEs of [12] (denoted as `FT21`) and `MMRC` [30] may be suboptimal. In this work, we aim to achieve the *exact*-optimal MSE under *both* communication and privacy constraints.

Furthermore, we show that `SQKR`, `FT21`, and `MMRC` can be viewed as special cases in our framework – i.e., (random) coding with their own codebook design. We elaborate on this in Section 5 and provide more details on prior work in Appendix I.

## 2 Problem Setting and Preliminaries

In this section, we formally define LDP (with shared randomness) and describe our problem setting.

**Local Differential Privacy (LDP)**  A randomized algorithm $\mathcal{Q} : \mathcal{X} \to \mathcal{Y}$ is $\varepsilon$-LDP if

$$\forall x, x' \in \mathcal{X}, y \in \mathcal{Y}, \frac{\mathcal{Q}(y|x)}{\mathcal{Q}(y|x')} \le e^{\varepsilon}. \tag{1}$$

**LDP with Shared Randomness**  In this work, we assume that the encoder and the decoder have access to a shared source of randomness $U \in \mathcal{U}$, where the random encoder (randomizer) $\mathcal{Q}$ privatizes $x$ with additional randomness $U$. Then, the corresponding $\varepsilon$-LDP constraint is

$$\forall x, x' \in \mathcal{X}, y \in \mathcal{Y}, \frac{\mathcal{Q}(y|x,u)}{\mathcal{Q}(y|x',u)} \le e^{\varepsilon} \tag{2}$$

for $P_U$-almost all $u$.

**Notation**  We let $\mathbb{S}^{d-1} = \{u \in \mathbb{R}^d : ||u||_2 = 1\}$ denote the unit sphere, $e_i \in \mathbb{R}^d$ the standard basis vectors for $i = 1, \ldots, d$, $\lfloor k \rfloor$ the greatest integer less than or equal to $k$, and $[M] = \{1, \ldots, M\}$.

**Problem Setting**  We consider the $\ell_2$ mean estimation problem with $n$ users where each user $i$ has a private unit vector $v_i \in \mathbb{S}^{d-1}$ for $1 \le i \le n$. The server wants to recover the mean $\frac{1}{n}\sum_{i=1}^n v_i$ after each user sends a message using up to $b$-bits under an $\varepsilon$-LDP constraint. We allow shared randomness between each user and the server. More concretely, the $i$-th user and the server both have access to a random variable $U_i \in \mathbb{R}^t$ (which is independent of the private local vector $v_i$) for some $t \ge 1$ and the $i$-th user has a random encoder (randomizer) $f_i : \mathbb{S}^{d-1} \times \mathbb{R}^t \to [M]$, where $M = 2^b$. We denote by $Q_{f_i}(m_i|v_i, u_i)$ the transition probability induced by the random encoder $f_i$, i.e., the probability that $f_i$ outputs $m_i$ given the source $v_i$ and the shared randomness $u_i$ is

$$\Pr[f_i(v_i, u_i) = m_i] = Q_{f_i}(m_i|v_i, u_i). \tag{3}$$

We require that the random encoder $f_i$ satisfies $\varepsilon$-LDP, i.e.,

$$\frac{Q_{f_i}(m_i|v_i, u_i)}{Q_{f_i}(m_i|v'_i, u_i)} \le e^{\varepsilon} \tag{4}$$

for all $v_i, v_i' \in \mathbb{S}^{d-1}$, $m_i \in [M]$ and $P_{U_i}$-almost all $u_i \in \mathbb{R}^t$.

The server receives $m_i = f_i(v_i, U_i)$ from all users and generates unbiased estimate of the mean $\mathcal{A}(m_1, \ldots, m_n, U_1, \ldots, U_n)$ that satisfies

$$\mathbb{E}\left[\mathcal{A}(m_1, \ldots, m_n, U_1, \ldots, U_n)\right] = \frac{1}{n}\sum_{i=1}^{n} v_i. \tag{5}$$

Then, the goal is to minimize the worst-case error

$$\mathrm{Err}_n(f, \mathcal{A}, P_{U^n}) = \sup_{v_1, \ldots, v_n \in \mathbb{S}^{d-1}} \mathbb{E}\left[\left\|\mathcal{A}(m_1, \ldots, m_n, U_1, \ldots, U_n) - \frac{1}{n}\sum_{i=1}^{n} v_i\right\|_2^2\right], \tag{6}$$

where $f$ denotes the collection of all encoders $(f_1, \ldots, f_n)$. We note that the error is also a function of the distribution of shared randomness, which was not the case for `PrivUnit` [2, 4].

## 3 Main Results

### 3.1 Canonical Protocols

Similar to Asi et al. [2], we first define the canonical protocol when both communication and privacy constraints exist. The canonical protocols are where the server recovers each user's vector and estimates the mean by averaging them. In other words, the server has a decoder $g_i : [M] \times \mathbb{R}^t \to \mathbb{S}^{d-1}$ for $1 \le i \le M$ which is dedicated to the $i$-th user's encoder $f_i$, where the mean estimation is a simple additive aggregation, i.e.,

$$\mathcal{A}^+(m_1, \ldots, m_n, U_1, \ldots, U_n) = \frac{1}{n}\sum_{i=1}^{n} g_i(m_i, U_i). \tag{7}$$

Our first result is that the *exact*-optimal mean estimation scheme should follow the above canonical protocol.

**Lemma 3.1.** *For any $n$-user mean estimation protocol $(f, \mathcal{A}, P_{U^n})$ that satisfies unbiasedness and $\varepsilon$-LDP, there exists an unbiased canonical protocol with decoders $g = (g_1, \ldots, g_n)$ that satisfies $\varepsilon$-LDP and achieves lower MSE, i.e.,*

$$\mathrm{Err}_n(f, \mathcal{A}, P_{U^n}) \ge \sup_{v_1, \ldots, v_n \in \mathbb{S}^{d-1}} \mathbb{E}\left[\left\|\frac{1}{n}\sum_{i=1}^{n} g_i(m_i, U_i) - \frac{1}{n}\sum_{i=1}^{n} v_i\right\|^2\right] \tag{8}$$

$$\ge \frac{1}{n^2}\sum_{i=1}^{n} \mathrm{Err}_1(f_i, g_i, P_{U_i}), \tag{9}$$

*where $\mathrm{Err}_1(f, g, P_U)$ is the worst-case error for a single user with a decoder $g$.*

The main proof techniques are similar to [2], where we define the marginalizing decoder:

$$g_i(m_i, U_i) = \mathbb{E}_{\{v_j, m_j, U_j\}_{j \ne i}}\left[n\mathcal{A}(\{m_j, U_j\}_{j=1}^{n}) \mid f_i(v_i, U_i) = m_i, U_i\right]. \tag{10}$$

The expectation in (10) is with respect to the uniform distribution of $v_j$'s. We defer the full proof to Appendix A.

Since the *exact*-optimal $n$-user mean estimation scheme is simply additively aggregating user-wise *exact*-optimal scheme, throughout the paper, we will focus on the single-user case and drop the index $i$ when it is clear from the context. In this simpler formulation, we want the server to have an unbiased estimate $\hat{v} = g(m, U)$, i.e.,

$$v = \mathbb{E}_{P_U, f}\left[g(f(v, U), U)\right] \tag{11}$$

$$= \mathbb{E}_{P_U}\left[\sum_{m=1}^{M} g(m, U)Q_f(m|v, U)\right] \tag{12}$$

for all $v \in \mathbb{S}^{d-1}$. We assume that the decoder $g : [M] \times \mathbb{R}^t \to \mathbb{R}^d$ is deterministic, since the randomized decoder does not improve the performance. Then, the corresponding error becomes

$$D(v, f, g, P_U) = \mathbb{E}_{P_U, f} \left[ \|g(f(U, v), U) - v\|^2 \right] \tag{13}$$

$$= \mathbb{E}_{P_U} \left[ \sum_{m=1}^{M} \|g(m, U) - v\|^2 Q_f(m|v, U) \right]. \tag{14}$$

Finally, we want to minimize the following worst-case error over all $(f, g)$ pairs that satisfy the unbiasedness condition in (12)

$$\mathrm{Err}_1(f, g, P_U) = \sup_{v \in \mathbb{S}^{d-1}} D(v, f, g, P_U). \tag{15}$$

## 3.2 Exact Optimality of the Codebook

We propose a special way of leveraging shared randomness, which we term as *random codebook*. First, we define a codebook $U^M = (U_1, \ldots, U_M) \in (\mathbb{R}^d)^M$, consisting of $M$ number of $d$-dimensional random vectors generated via shared randomness (i.e., both the server and the user know these random vectors). We then define the corresponding simple selecting decoder $g^+ : [M] \times (\mathbb{R}^d)^M \to \mathbb{R}^d$, which simply picks the $m$-th vector of the codebook upon receiving the message $m$ from the user:

$$g^+(m, U^M) = U_m. \tag{16}$$

Our first theorem shows that there exists a scheme with a random codebook and a simple selecting decoder that achieves the *exact*-optimal error. More precisely, instead of considering the general class of shared randomness (with general dimension $t$) and the decoder, it is enough to consider the random codebook $U^M \in (\mathbb{R}^d)^M$ as the shared randomness and the simple selector $g^+$ as the decoder.

**Lemma 3.2.** *For any $f, g, P_U$ with $U \in \mathbb{R}^t$ that are unbiased and that satisfy $\varepsilon$-LDP, there exists a shared randomness $\tilde{U}^M \in (\mathbb{R}^d)^M$ and random encoder $f_0 : \mathbb{S}^{d-1} \times (\mathbb{R}^d)^M \to [M]$ such that*

$$D(v, f, g, P_U) = D(v, f_0, g^+, P_{\tilde{U}^M}) \tag{17}$$

*for all $v \in \mathbb{S}^{d-1}$, where $f_0, g^+, P_{\tilde{u}^M}$ also satisfy unbiasedness and $\varepsilon$-LDP.*

The main step of the proof is to set an implicit random codebook with codewords $\tilde{U}_m = g(m, U)$ for $m = 1, \ldots, 2^b$ and show that we can obtain an essentially equivalent scheme with a different form of shared randomness $\tilde{U}^M$, which is an explicit random codebook. The detailed proof is given in Appendix B. Thus, without loss of generality, we can assume $t = M \times d$ and the random codebook $U^M$ is the new shared randomness, where the decoder is a simple selector. Since we fix the decoder, we drop $g$ to simplify our notation. We say the random encoder $f$ satisfies unbiasedness condition if

$$\mathbb{E}_{P_U} \left[ \sum_{m=1}^{M} U_m Q_f(m|v, U^M) \right] = v, \tag{18}$$

and the worst-case error is

$$\mathrm{Err}(f, P_{U^M}) = \sup_{v \in \mathbb{S}^{d-1}} D(v, f, P_{U^M}) \tag{19}$$

$$= \sup_{v \in \mathbb{S}^{d-1}} \mathbb{E}_{P_{U^M}} \left[ \sum_{m=1}^{M} \|U_m - v\|^2 Q_f(m|v, U) \right]. \tag{20}$$

Thus, the goal is now to find the *exact*-optimum codebook generating distribution $P_{U^M}$, and the random encoder $f$ (or the probability assignment $Q_f(\cdot|v, U)$). We then argue that the *exact*-optimal codebook should be rotationally symmetric.

**Definition 3.3.** *A random codebook $U^M \in (\mathbb{R}^d)^M$ is **rotationally symmetric** if $(U_1, \ldots, U_M) \overset{(d)}{=} (A_0 U_1, \ldots, A_0 U_M)$ for any $d \times d$ orthonormal matrix $A_0$.*

The next lemma shows that the *exact*-optimal $P_{U^M}$ is rotationally symmetric.

**Lemma 3.4.** *Let $P_{U^M}$ be a codebook generating distribution, and suppose random encoder $f$ satisfies unbiasedness and $\varepsilon$-LDP. Then, there exists a random encoder $f_1$ and rotationally symmetric random codebook $\bar{U}^M$ such that*

$$\mathrm{Err}(f, P_{U^M}) \geq \mathrm{Err}(f_1, P_{\bar{U}^M}), \tag{21}$$

*which also satisfies unbiasedness and $\varepsilon$-LDP.*

This is mainly because the goal is to minimize the worst-case error, and the codebook-generating distribution should be symmetric in all directions. The proof is provided in Appendix C. The next lemma shows that the *exact*-optimal scheme has constant error for all $v \in \mathbb{S}^{d-1}$.

**Lemma 3.5.** *For any rotationally symmetric codebook generating distribution $P_{U^M}$ and an unbiased randomized encoder $f$ that satisfies $\varepsilon$-LDP, there exists a random encoder $f_2$ such that*

$$\text{Err}(f, P_{U^M}) \geq \text{Err}(f_2, P_{U^M}), \text{ where } D(v, f_2, P_{U^M}) = D(v', f_2, P_{U^M}) \tag{22}$$

*for all $v, v' \in \mathbb{S}^{d-1}$.*

The formal proof is given in Appendix D. Since the codebook is symmetric (Lemma 3.4), the *exact*-optimal encoding strategy remains the same for any input $v$. Thus, without loss of generality, we can assume that the input is a standard unit vector $v = e_1 = (1, 0, \ldots, 0)$.

### 3.3 Rotationally Symmetric Simplex Codebook

Now, we focus on a particular rotationally symmetric codebook. Notice that the codebook $U^M$ has a similar role to the codebook in lossy compression, in the sense that we prefer the codeword $U_m$ close to the input vector $v$. Thus, it is natural to consider the maximally separated codebook so that the $M$ vectors $U_1, \ldots, U_M$ cover the source space effectively. For $M < d$, the maximally separated $M$ vectors on the unit sphere $\mathbb{S}^{d-1}$ is a simplex. More precisely, let $s_1, \ldots, s_M \in \mathbb{R}^d$ form a simplex:

$$(s_i)_j = \begin{cases} \frac{M-1}{\sqrt{M(M-1)}} & \text{if } i = j \\ -\frac{1}{\sqrt{M(M-1)}} & \text{if } i \neq j \text{ and } j \leq M \\ 0 & \text{if } j > M \end{cases} \tag{23}$$

Then, we can define the rotationally symmetric simplex codebook $U^M$

$$(U_1, U_2, \ldots, U_M) \stackrel{(d)}{=} (rAs_1, rAs_2, \ldots, rAs_M), \tag{24}$$

where $A$ is uniformly drawn orthogonal matrix and $r > 0$ is a normalizing constant. We then need to find the corresponding encoder $f$ that minimizes the error. Recall that the error is

$$\mathbb{E}_{P_{U^M}} \left[ \sum_{m=1}^{M} \|U_m - v\|^2 Q_f(m|v, U) \right], \tag{25}$$

and it is natural to assign high probabilities to the message $m$ with low distortion $\|U_m - v\|^2$ as long as $\varepsilon$-LDP constraint allows. More precisely, we call the following probability assignment "$k$-closest" encoding:

$$Q_f(m|v, U^M) = \begin{cases} \frac{e^\varepsilon}{ke^\varepsilon + (M-k)} & \text{if } \|v - U_m\|^2 \text{ is one of the } \lfloor k \rfloor \text{ smallest} \\ \frac{(k - \lfloor k \rfloor)(e^\varepsilon - 1) + 1}{ke^\varepsilon + (M-k)} & \text{if } \|v - U_m\|^2 \text{ is the } \lfloor k \rfloor + 1\text{-th smallest }, \\ \frac{1}{ke^\varepsilon + (M-k)} & \text{otherwise} \end{cases} \tag{26}$$

where we allow non-integer $k$. The choice of $r = r_k$ is described in Section 3.4. We call this approach Randomly Rotating Simplex Coding (RRSC) and provide the pseudocode in Algorithm 1. We note that the codewords $U_m$'s with smallest $\|v - U_m\|^2$ and codewords $U_m$'s with largest $\langle v, U_m \rangle$ coincide for a codebook with fixed-norm codewords $U_m$'s, which is the case for the rotationally symmetric simplex codebook. Our main theorem is that the $k$-closest encoding is *exact*-optimum if the codebook generating distribution is rotationally symmetric simplex.

**Theorem 3.6.** *For a rotationally symmetric simplex codebook, there exists a $k$ such that the "$k$-closest" encoding is the exact-optimum unbiased scheme that satisfies $\varepsilon$-LDP constraint.*

The main step of the proof is to show that all the probabilities should be either the maximum or the minimum in order to minimize the error, and the proof is given in Appendix E.

### 3.4 $k$-closest Encoding for General Rotationally Symmetric Codebook

In this section, we demonstrate that the $k$-closest encoding consistently yields an unbiased scheme for any rotationally symmetric codebook. To be more specific, for any given spherically symmetric

**Algorithm 1** Randomly Rotating Simplex Coding $\texttt{RRSC}(k)$.

---

**Inputs:** $v \in \mathbb{S}^{d-1}$, $k$, $r_k$, codebook size $M = 2^b$.
    **Codebook Generation:**
    Generate the simplex $s_1, \ldots, s_M \in \mathbb{R}^d$ in (23).
    Sample orthogonal matrix $A \in \mathbb{R}^{d \times d}$ uniformly using the shared random $\texttt{SEED}$.
    Generate the codebook $U^M$: $(U_1, U_2, \ldots, U_M) \leftarrow (r_k A s_1, r_k A s_2, \ldots, r_k A s_M)$.
    **Encoding:**
    **for** $m \in [M]$ **do**
        **if** $\langle v, U_m \rangle$ is one of the $k$ largest **then**
            $Q_f(m|v, U^M) \leftarrow \frac{e^\varepsilon}{ke^\varepsilon + (M-k)}$
        **else**
            $Q_f(m|v, U^M) \leftarrow \frac{1}{ke^\varepsilon + (M-k)}$
        **end if**
    **end for**
    Sample codeword index $m^* \leftarrow Q_f(\cdot|v, U^M)$.
**Output:** $m^*$, encoded in $b = \log M$ bits.

---

codebook $U^M$, there exists a scalar $r_k$ that ensures that the $k$-closest encoding with $r_k U^M = (r_k U_1, \ldots, r_k U_M)$ is unbiased. Let $T_k(v, U^M) = \{m : U_m \text{ is one of the } k\text{-closest}\}$, and without loss of generality, let us assume $v = e_1$. Then,

$$\mathbb{E}_{P_{U^M}} \left[ \sum_{m=1}^M Q_f(m|e_1, U^M) U_m \right]$$

$$= \mathbb{E}_{P_{U^M}} \left[ \frac{e^\varepsilon - 1}{ke^\varepsilon + (M-k)} \sum_{m \in T_k(e_1, U^M)} U_m + \frac{1}{ke^\varepsilon + (M-k)} \sum_{m=1}^M U_m \right] \quad (27)$$

$$= \mathbb{E}_{P_{U^M}} \left[ \frac{e^\varepsilon - 1}{ke^\varepsilon + (M-k)} \sum_{m \in T_k(e_1, U^M)} U_m \right], \quad (28)$$

where $\mathbb{E}\left[ \sum U_m \right] = 0$ due to rotationally symmetric codebook and we assume an integer $k$ for the sake of simplicity. Since the codebook is rotationally symmetric and we pick $k$-closest vectors toward $v = e_1$, each codeword $U_m \in T_k(e_1, U^M)$ is symmetric in all directions other than $v = e_1$. Thus, in expectation, the decoded vector is aligned with $e_1$, and there exists $r_k$ such that

$$r_k \times \mathbb{E}_{P_{U^M}} \left[ \sum_{m=1}^M Q_f(m|e_1, U^M) U_m \right] = e_1. \quad (29)$$

For a rotationally symmetric simplex codebook, where $U_m = A s_m$ for a uniform random orthogonal matrix $A$, we have an (almost) analytic formula.

**Lemma 3.7.** *Normalization constant $r_k$ for* $\texttt{RRSC}(k)$ *is*

$$r_k = \frac{ke^\varepsilon + M - k}{e^\varepsilon - 1} \sqrt{\frac{M-1}{M}} \frac{1}{C_k}, \quad (30)$$

*where $C_k{}^2$ is an expected sum of top-$k$ coordinates of uniform random vector $a \in \mathbb{S}^{d-1}$.*

The key idea in the proof is to show that encoding $e_1$ with $As^M$ is equivalent to encoding uniform random vector $a \in \mathbb{S}^{d-1}$ with $s^M$. The formal proof is provided in Appendix F.

The following lemma controls the asymptotic behavior of $C_k$:

**Lemma 3.8.** *Let $C_k$ be defined as in Lemma 3.7. Then, it holds that*

$$C_k = O\left( \sqrt{\frac{k^2 \log M}{d}} \right). \quad (31)$$

---

[2]Note that $C_k$ depends on $k, d$, and $M$, but for ease of presentation, we suppress the dependency on $d$ and $m$ here and only present the full expression in the proof.

In addition, there exist absolute constants $C_1, C_2 > 0$ such that as long as $\lfloor M/k \rfloor > C_1$ and $k > C_2$,

$$C_k = \Omega\left(\sqrt{\frac{k^2}{d}\log\left(\frac{M}{k}\right)}\right). \tag{32}$$

As a corollary, Lemma 3.8 implies the order-wise optimality of `RRSC`:

$$\mathsf{Err}(\mathtt{RRSC}) \leq r_k^2 - 1 = O\left(\frac{\left(e^\varepsilon - 1 - \frac{M}{k}\right)^2}{(e^\varepsilon - 1)^2} \cdot \frac{d}{\log\left(\frac{M}{k}\right)}\right).$$

By picking $k = \max\left(1, Me^{-\varepsilon}\right)$, the above error is $O\left(\frac{d}{\min(\varepsilon^2, \varepsilon, b)}\right)$. We provide the proof of Lemma 3.8 in Appendix G.

### 3.5  Convergence to PrivUnit

As the communication constraint $b$ increases, the *exact*-optimal scheme with communication constraint should coincide with the *exact*-optimal scheme *without* communication constraint, which is `PrivUnit`. Note that the rotationally symmetric simplex can be defined only when $M = 2^b < d$, due to its simplex structure. However, we have a natural extension where the codebook is a collection of $M$ (nearly) maximally separated vectors on the sphere of radius $r$, where we can assume that $M$ codewords are uniformly distributed on the sphere of radius $r_k$ if $M$ is large enough. Consider the case where $q = \frac{k}{M}$ is fixed and $M = 2^b$ is large. Since the $k$-closest encoding yields an unbiased scheme with error $\mathsf{Err}(f, P_{U^M}) = r_k^2 - 1$, where $r_k$ is normalizing constant, for uniformly distributed $M$ codewords on the sphere, the constant $r_k$ should satisfy

$$r_k \times \frac{e^\varepsilon - 1}{ke^\varepsilon + (M - k)}\mathbb{E}\left[\sum_{m \in \text{top-}k} U_{m,1}\right] = 1 \tag{33}$$

where $U_{m,1}$ is the first coordinate of uniformly drawn $U_m$ from the unit sphere $\mathbb{S}^{d-1}$. Then, as $M$ increases, $U_{m,1}$ being one of the top-$k$ becomes equivalent to $U_{m,1} > \gamma$, where $\gamma$ is the threshold such that $\Pr[U_{m,1} > \gamma] = q$. Hence, assigning higher probabilities to the top-$k$ closest codewords becomes equivalent to assigning high probabilities to the codewords with $\langle U_m, e_1 \rangle > \gamma$ where $v = e_1$. This is essentially how `PrivUnit` operates.

### 3.6  Complexity of `RRSC`

Each user has $d \times d$ orthonormal matrix $A$ and needs to find $k$ smallest $\langle v, As_m \rangle$ for $1 \leq m \leq M$. Since $\langle v, As_m \rangle = \langle A^\mathsf{T}v, s_m \rangle$, it requires $O(d^2)$ to compute $A^\mathsf{T}v$ and additional $O(Md)$ to compute all inner products for $1 \leq m \leq M$. However, if $M \ll d$, we have a simpler equivalent scheme using

$$\langle A^\mathsf{T}v, s_m \rangle = \sqrt{\frac{M}{M - 1}}a_m^\mathsf{T}v - \sum_{i=1}^{M} a_i^\mathsf{T}v\frac{1}{\sqrt{M(M - 1)}}, \tag{34}$$

where $a_m^\mathsf{T}$ is the $m$-th row of the matrix $A$. Then, it only requires storing the first $M$ rows of the matrix and $O(Md)$ to obtain all inner products in (34) by avoiding $O(d^2)$ to construct $A^\mathsf{T}v$.

On the other hand, the server computes $As_m$ upon receiving a message $m$. The corresponding time complexity is $O(Md)$ (per user) since $s_m$ has $M$ non-zero values. We note that both `MMRC` [30] and `FT21` [12] require the same encoding complexity $O(Md)$ as `RRSC`, where they choose $M = O(\exp(\varepsilon))$.

## 4  Experiments

We empirically demonstrate the communication-privacy-utility tradeoffs of `RRSC` and compare it with *order*-optimal schemes under privacy and communication constraints, namely `SQKR` [6] and `MMRC` [30]. We also show that `RRSC` performs comparably with `PrivUnit` [4], which offers the *exact*-optimal privacy-utility tradeoffs without communication constraints [2]. In our simulations, we use

the "optimized" `PrivUnit` mechanism, called `PrivUnitG`, introduced in [2], which performs better than `PrivUnit` in practice since it provides an easy-to-analyze approximation of `PrivUnit` but with analytically better-optimized hyperparameters. Similar to [6, 30], we generate data independently but non-identically to capture the distribution-free setting with $\mu \neq 0$. More precisely, for the first half of the users, we set $v_1, \ldots, v_{n/2} \overset{\text{i.i.d}}{\sim} N(1,1)^{\otimes d}$; and for the second half of the users, we set $v_{n/2+1}, \ldots, v_n \overset{\text{i.i.d}}{\sim} N(10,1)^{\otimes d}$. We further normalize each $v_i$ to ensure that they lie on $\mathbb{S}^{d-1}$. We report the average $\ell_2$ error over 10 rounds together with the confidence intervals. To find the optimal values for $k$ and $r_k$, we compute the optimal $r_k$ using the formula in (33) for $k = 1, \ldots, M$ and pick the $k$ that gives the smallest $r_k$ (which corresponds to the bias). To estimate the expectation $C_k$ in (33), we run a Monte Carlo simulation with $1M$ trials. We report the $k$ we use for each experiment in the captions. Additional experimental results are provided in Appendix H.

In Figure 1-(left, middle), we report $\ell_2$ error for $\varepsilon = 1, \ldots, 8$, where for each method (except `PrivUnitG`), the number of bits is equal to $b = \varepsilon$. In Figure 1-(right), we report $\ell_2$ error by fixing $\epsilon = 6$ and sweeping the bitrate from $b = 1$ to $b = 8$ for `RRSC` and `MMRC`. For `SQKR`, we only sweep for $b \leq \varepsilon$ as it leads to poor performance for $b > \varepsilon$. In each figure, `RRSC` performs comparably to `PrivUnitG` even for small $b$ and outperforms both `SKQR` and `MMRC` by large margins.

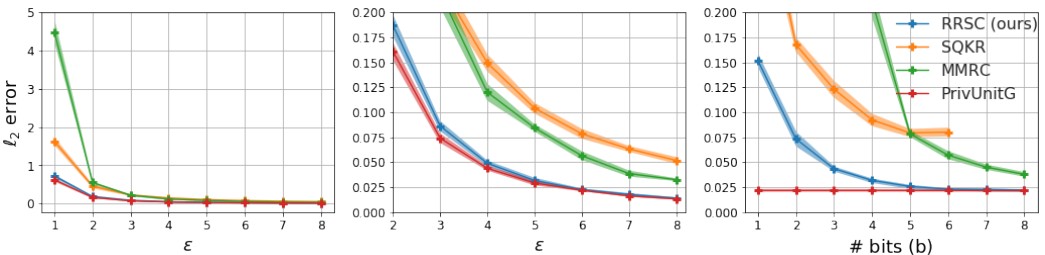

Figure 1: Comparison of `RRSC` with `SQKR` [6], `MMRC` [30], and `PrivUnitG` [2]. **(left)** $\ell_2$ error vs $\varepsilon$ with $n = 5000$, $d = 500$. The number of bits is $b = \varepsilon$ for `RRSC`, `SQKR`, and `MMRC`. The choice of $k$ for $k$-closest encoding is $k = 1$ for each $\varepsilon$. **(middle)** Same plot zoomed into higher $\varepsilon$, lower $\ell_2$ error region. **(right)** $\ell_2$ error vs number of bits $b$ for $n = 5000$, $d = 500$, and $\varepsilon = 6$. For `SQKR`, we only report $b \leq \varepsilon = 6$ since it performs poorly when $b > \varepsilon$. The choice of $k$ for $k$-closest encoding is $k = [1, 1, 1, 1, 1, 1, 2, 4]$ for $b = [1, 2, 3, 4, 5, 6, 7, 8]$, respectively.

The codebase for this work is open-sourced at `https://github.com/BerivanIsik/rrsc`.

## 5 Discussion & Conclusion

We proved that using a rotationally symmetric codebook is a necessary condition for the *exact* optimality of mean estimation mechanisms with privacy and communication constraints. We then proposed Random Rotating Simplex Coding (`RRSC`) based on a $k$-closest encoding mechanism and proved that `RRSC` is *exact*-optimal for the random rotating simplex codebook. We now discuss some important features of `RRSC` and provide conjectures for future work.

**Unified Framework** It turns out that `SQKR` [6], `FT21` [12] and `MMRC` [30] can be viewed as special cases in our framework. Specifically, `SQKR` [6] uses Kashin's representation of $v = \sum_{j=1}^{N} a_j u_j$, where $\{a_j\}_{j=1}^{N} \in [-c/\sqrt{d}, c/\sqrt{d}]$ for some $(1 + \mu)d$ with $\mu > 0$ and $c > 0$. Then the `SQKR` encoder quantizes each $a_j$ into a 1-bit message $q_i$, and draws $k$ samples with the help of shared randomness. This can be viewed as random coding with a codebook-generating distribution. More concretely, the corresponding codebook $U^M$ consists of $k$ non-zero values of $\pm c/\sqrt{d}$ where the randomness is from selecting $k$ indices using shared randomness. On the other hand, since `MMRC` [30] is simulating the channel corresponding to a privacy mechanism, it can be viewed as pre-generating random codewords $U^M$ according to the reference distribution, where the importance sampling is also a way of assigning probabilities to each codeword. As elaborated in Section 3.5, it is observed that with an increase in the communication constraint $b$, the suggested $k$-closest encoding gradually transforms into a threshold-based encoding, analogous to that of `MMRC`. The codebook associated with `FT21` [12]

depends on the PRG it uses. Let $\mathsf{PRG} : \{0,1\}^b \to \{0,1\}^{\Theta(d)}$ be a PRG that takes a $b$-bit seed and maps it into $\Theta(d)$ bits, where $b \ll d$, and let $g : \{0,1\}^{\Theta(d)} \to \mathbb{R}^d$. For example, if we represent each coordinate of $x \in \mathbb{R}^d$ as a 32-bit float, then $g(\cdot)$ maps the float representation of $x$ (a 32-bit string) to $x$. With a PRG, FT21 mimics `PrivUnit` by first generating a $b$-bit seed $m$, computing $g(\mathsf{PRG}(m))$, and then performing rejection sampling on the seed space. The above procedure can be treated as a special case in our framework, where the *deterministic* codebook consists of $2^b$ points on $\mathbb{R}^d$: $\mathcal{C}_{\mathsf{FT21}} := \{g(\mathsf{PRG}(m)) : m \in \{0,1\}^b\}$. The probabilities assigned to each codeword according to the rejection sampling are equivalent to a threshold-based assignment.

**Shared randomness**  When $M \leq d + 1$, additional randomization is required during codebook generation to achieve an unbiased scheme, as discussed in [13]. Furthermore, both the encoder and decoder must possess this randomization information. In the proposed `RRSC` scheme, this randomization is achieved through the random rotation of the simplex code using shared randomness. However, it is possible to circumvent the need for shared randomness by having the server generate random rotation matrices using its private coin and communicate them to the users. This approach replaces shared randomness with downlink communication, which is typically more affordable than uplink communication. It should be noted that directly transmitting the rotation matrices would require $O(d^2)$ bits. Nonetheless, the server can generate them using a predetermined pseudo-random generator (PRG) and transmit only the seeds of it to the users. Drawing from a similar argument as in [12], assuming the existence of exponentially strong PRGs, seeds with $\mathsf{polylog}(d)$ bits are sufficient.

**Future Work**  We showed the *exact*-optimality of $k$-closest encoding for the rotating simplex codebook. In general, it also achieves unbiasedness and the following error formulation $\mathbb{E}_{P_{U^M}} \left[ \sum_{m=1}^{M} Q_f(m|v, U^M) \|v - U_m\|^2 \right]$ implies the *exact*-optimality of $k$-closest encoding for any rotationally symmetric codebook, which leads us to the following conjecture.

**Conjecture 5.1.** *The proposed $k$-closest encoding is exact-optimal for any rotationally symmetric codebook.*

It also remains unclear whether $k$ can depend on the realization of the codebook $U^M$ in general, which we leave to future work. We also proved that the *exact*-optimal codebook must be rotationally symmetric. We conjecture that the maximally separated codebook (simplex codebook) is *exact*-optimal as it provides the most effective coverage of the space $\mathbb{S}^{d-1}$. This, too, is left as a topic for future work.

**Conjecture 5.2.** *The rotationally symmetric simplex codebook is the exact-optimal codebook.*

**Limitations and Broader Impact**  While we take an important step towards exact optimality by proving several necessary conditions and by providing a mechanism that is *exact*-optimal for a family of codebooks, we still have the above conjectures left to be proven in future work.

# 6   Acknowledgement

The authors would like to thank Peter Kairouz for the helpful discussions in the early stage of this work; and Hilal Asi, Vitaly Feldman, and Kunal Talwar for sharing their implementation for the `PrivUnitG` mechanism [2]. BI was supported by a Stanford Graduate Fellowship, a Google PhD Fellowship, and a Meta research grant. WNC was supported by NSF under Grant CCF-2213223. AN was supported by the Basic Science Research Program through the National Research Foundation of Korea (NRF) funded by the Ministry of Education (2021R1F1A1059567).

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

# A  Proof of Lemma 3.1

*Proof.* For $n$-user mean estimation protocol $(f, \mathcal{A}, P_{UM})$, following the notation and steps from [2, Proof of Lemma 3.1], we define the marginalized output

$$\tilde{g}_i(m_i, U_i; v^n) = \mathbb{E}_{\{m_j, U_j\}_{j \neq i}} \left[ n\mathcal{A}(\{m_j, U_j\}_{j=1}^n) \mid f_i(v_i, U_i) = m_i, U_i, v^{n \setminus i} \right]. \tag{35}$$

Then, we define the user-specific decoder by averaging $g_i(m_i, U_i; v^n)$ with respect to i.i.d. uniform $P_{\text{unif}}$:

$$g_i(m_i, U_i) = \mathbb{E}_{v^{n \setminus i} \sim P_{\text{unif}}} \left[ \tilde{g}_i(m_i, U_i; v^n) \right] \tag{36}$$

where $v^{n \setminus i}$ indicates the $v^n$ vector except $v_i$. Due to the symmetry of $P_{\text{unif}}$, it is clear that $g_i$ is unbiased. We also define

$$\hat{\mathcal{R}}_{\leq i}(\{v_j, m_j, U_j\}_{j=1}^i) = \mathbb{E}_{v_j \sim P_{\text{unif}}, j > i} \left[ n\mathcal{A}(\{m_j, U_j\}_{j=1}^n) - \sum_{j=1}^i v_j \mid \{v_j, m_j, U_j\}_{j=1}^i \right] \tag{37}$$

Consider an average error where $v_1, \ldots, v_n$ are drawn i.i.d. uniformly on the sphere $\mathbb{S}^{d-1}$.

$$\mathbb{E}_{\{v_j, m_j, U_j\}_{j=1}^n} \left[ \left\| n\mathcal{A}(\{m_j, U_j\}_{j=1}^n) - \sum_{j=1}^n v_j \right\|^2 \right]$$

$$= \mathbb{E}_{\{v_j, m_j, U_j\}_{j=1}^n} \left[ \left\| \hat{\mathcal{R}}_{\leq n}(\{v_j, m_j, U_j\}_{j=1}^n) \right\|^2 \right] \tag{38}$$

$$= \mathbb{E}_{\{v_j, m_j, U_j\}_{j=1}^n} \left[ \left\| \hat{\mathcal{R}}_{\leq n}(\{v_j, m_j, U_j\}_{j=1}^n) - \hat{\mathcal{R}}_{\leq n-1}(\{v_j, m_j, U_j\}_{j=1}^{n-1}) + \hat{\mathcal{R}}_{\leq n-1}(\{v_j, m_j, U_j\}_{j=1}^{n-1}) \right\|^2 \right] \tag{39}$$

$$= \mathbb{E}_{\{v_j, m_j, U_j\}_{j=1}^n} \left[ \left\| \hat{\mathcal{R}}_{\leq n}(\{v_j, m_j, U_j\}_{j=1}^n) - \hat{\mathcal{R}}_{\leq n-1}(\{v_j, m_j, U_j\}_{j=1}^{n-1}) \right\|^2 \right]$$

$$\quad + \mathbb{E}_{\{v_j, m_j, U_j\}_{j=1}^{n-1}} \left[ \left\| \hat{\mathcal{R}}_{\leq n-1}(\{v_j, m_j, U_j\}_{j=1}^{n-1}) \right\|^2 \right] \tag{40}$$

$$= \sum_{i=1}^n \mathbb{E}_{\{v_j, m_j, U_j\}_{j=1}^i} \left[ \left\| \hat{\mathcal{R}}_{\leq i}(\{v_j, m_j, U_j\}_{i=1}^n) - \hat{\mathcal{R}}_{\leq i-1}(\{v_j, m_j, U_j\}_{j=1}^{i-1}) \right\|^2 \right] \tag{41}$$

$$\geq \sum_{i=1}^n \mathbb{E}_{m_i, U_i} \left[ \left\| \mathbb{E}_{\{v_j, m_j, U_j\}_{j=1}^{i-1}} \left[ \hat{\mathcal{R}}_{\leq i}(\{v_j, m_j, U_j\}_{i=1}^n) - \hat{\mathcal{R}}_{\leq i-1}(\{v_j, m_j, U_j\}_{j=1}^{i-1}) \right] \right\|^2 \right] \tag{42}$$

$$= \sum_{i=1}^n \mathbb{E}_{m_i, U_i} \left[ \|g_i(m_i, U_i) - v_i\|^2 \right]. \tag{43}$$

Then, we need to show the same inequality for the worst-case error.

$$\sup_{v_1, \ldots, v_n} \mathbb{E}_{\{m_j, U_j\}_{j=1}^n} \left[ \left\| n\mathcal{A}(\{m_j, U_j\}_{j=1}^n) - \sum_{j=1}^n v_j \right\|^2 \right]$$

$$\geq \mathbb{E}_{\{v_j, m_j, U_j\}_{j=1}^n} \left[ \left\| n\mathcal{A}(\{m_j, U_j\}_{j=1}^n) - \sum_{j=1}^n v_j \right\|^2 \right] \tag{44}$$

$$= \sum_{i=1}^n \mathbb{E}_{v_i, m_i, U_i} \left[ \|g_i(m_i, U_i) - v_i\|^2 \right] \tag{45}$$

$$= \sum_{i=1}^n \sup_{v_i} \mathbb{E}_{m_i, U_i} \left[ \|g_i(m_i, U_i) - v_i\|^2 \right] \tag{46}$$

where the last equality (46) is from Lemma 3.2, Lemma 3.4, and Lemma 3.5. Thus, the user-specific decoder achieves lower MSE:

$$\mathrm{Err}_n(f, \mathcal{A}, P_{U^n}) \geq \frac{1}{n} \sum_{i=1}^{n} \mathrm{Err}_1(f_i, g_i, P_{U_i}). \tag{47}$$

Since we keep random encoder $f_i$ the same, the canonical protocol with $g_i$ also satisfies $\varepsilon$-LDP constraint. This concludes the proof. $\qquad\square$

## B   Proof of Lemma 3.2

*Proof.* Let $\tilde{U}_m = g(m, U)$ for all $1 \leq m \leq M$. Without loss of generality $g(\cdot, U)$ is one-to-one, i.e., $\{u : \tilde{u}_m = g(m, u)$ for all $m\}$ has at most one element (with probability 1), and $u = g^{-1}(\tilde{u}^M)$ is well-defined. Then, we define a randomizer $f_0(v, \tilde{U}^M)$ that satisfies

$$Q_{f_0}(m|v, \tilde{u}^M) = Q_f(m|v, g^{-1}(\tilde{u}^M)). \tag{48}$$

It is clear that $f_0$ satisfies $\varepsilon$-LDP constraint. Then,

$$D(v, f_0, g^+, P_{\tilde{U}^M}) = \mathbb{E}_{f_0, P_{\tilde{U}^M}} \left[ \|g^+(f_0(v, \tilde{U}^M), \tilde{U}^M) - v\|^2 \right] \tag{49}$$

$$= \mathbb{E}_{P_{\tilde{U}^M}} \left[ \sum_{m=1}^{M} Q_{f_0}(m|v, \tilde{U}^M) \|\tilde{U}_m - v\|^2 \right] \tag{50}$$

$$= \mathbb{E}_{f, P_U} \left[ \sum_{m=1}^{M} Q_f(m|v, U) \|g(m, U) - v\|^2 \right] \tag{51}$$

$$= \mathbb{E}_{f, P_U} \left[ \|g(f(v, U), U) - v\|^2 \right] \tag{52}$$

$$= D(v, f, g, P_U). \tag{53}$$

We also need to show that the composition of the new randomizer $f_0$ and selector $g^+$ is unbiased.

$$\mathbb{E}_{P_{\tilde{U}^M}} \left[ g^+(f_0(v, \tilde{U}^M), \tilde{U}^M) \right] = \mathbb{E}_{f_0, P_{\tilde{U}^M}} \left[ \sum_{m=1}^{M} Q_{f_0}(m|v, \tilde{U}^M) \tilde{U}_m \right] \tag{54}$$

$$= \mathbb{E}_{f, P_U} \left[ \sum_{m=1}^{M} Q_f(m|v, U) g(m, U) \right] \tag{55}$$

$$= \mathbb{E}_{f, P_U} \left[ g(f(v, U), U) \right] \tag{56}$$

$$= v. \tag{57}$$

Finally, $Q_{f_0}(m|v, \tilde{u}^M)$ is a valid transition probability, since

$$\sum_{m=1}^{M} Q_{f_0}(m|v, \tilde{u}^M) = \sum_{m=1}^{M} Q_f(m|v, g^{-1}(\tilde{u}^M)) = 1 \tag{58}$$

for all $\tilde{u}^M$. This concludes the proof. $\qquad\square$

## C   Proof of Lemma 3.4

*Proof.* Let $A$ be a uniformly random orthogonal matrix and $\bar{U}^M = A^{\mathsf{T}} U^M$. We further let $f_1$ be a randomized encoder that satisfies

$$Q_{f_1}(m|v, \bar{U}^M) = \mathbb{E}_A \left[ Q_f(m|Av, A\bar{U}^M) | \bar{U}^M \right]. \tag{59}$$

Then, $Q_{f_1}$ is a valid probability since

$$\sum_{m=1}^{M} Q_{f_1}(m|v, \bar{U}^M) = \mathbb{E}_A \left[ \sum_{m=1}^{M} Q_f(m|Av, A\bar{U}^M) | \bar{U}^M \right] = 1. \tag{60}$$

Also, we have

$$\frac{Q_{f_1}(m|v, \bar{U}^M)}{Q_{f_1}(m|v', \bar{U}^M)} = \frac{\mathbb{E}_A \left[ Q_f(m|Av, A\bar{U}^M) | \bar{U}^M \right]}{\mathbb{E}_A \left[ Q_f(m|Av', A\bar{U}^M) | \bar{U}^M \right]} \tag{61}$$

$$\leq \frac{\mathbb{E}_A \left[ e^\varepsilon Q_f(m|Av', A\bar{U}^M)|\bar{U}^M \right]}{\mathbb{E}_A \left[ Q_f(m|Av', A\bar{U}^M)|\bar{U}^M \right]} \tag{62}$$

$$= e^\varepsilon. \tag{63}$$

Finally, we need to check unbiasedness.

$$\mathbb{E}_{P_{\bar{U}M}} \left[ Q_{f_1}(m|v, \bar{U}^M)\bar{U}_m \right] = \mathbb{E}_{A, P_{UM}} \left[ \sum_{m=1}^M Q_f(m|Av, A\bar{U}^M)\bar{U}_m \right] \tag{64}$$

$$= \mathbb{E}_{A, P_{UM}} \left[ \sum_{m=1}^M Q_f(m|Av, U^M)A^\intercal U_m \right] \tag{65}$$

$$= \mathbb{E}_A \left[ A^\intercal \mathbb{E}_{P_{UM}} \left[ \sum_{m=1}^M Q_f(m|Av, U^M)U_m \right] \right] \tag{66}$$

$$= \mathbb{E}_A \left[ A^\intercal Av \right] \tag{67}$$

$$= v. \tag{68}$$

The key step is that the original encoder $f$ is unbiased, which implies

$$\mathbb{E}_{P_{UM}} \left[ \sum_{m=1}^M Q_f(m|Av, U^M)U_m \right] = Av \tag{69}$$

for all $A$.

Now, we are ready to prove the main inequality.

$$\mathrm{Err}(f, P_{UM}) = \sup_v D(v, f, P_{UM}) \tag{70}$$

$$\geq \mathbb{E}_A \left[ D(Av, f, P_{UM}) \right] \tag{71}$$

$$= \mathbb{E}_A \left[ \mathbb{E}_{P_{UM}} \left[ \sum_{m=1}^M Q_f(m|Av, U^M)\|U_m - Av\|^2 \right] \right] \tag{72}$$

$$= \mathbb{E}_{P_{UM}, A} \left[ \sum_{m=1}^M Q_f(m|Av, A\bar{U}^M)\|\bar{U}_m - v\|^2 \right] \tag{73}$$

$$= \mathbb{E}_{P_{\bar{U}M}} \left[ \sum_{m=1}^M \mathbb{E}_A \left[ Q_f(m|Av, A\bar{U}^M)|\bar{U}^M \right] \|\bar{U}_m - v\|^2 \right] \tag{74}$$

$$= \mathbb{E}_{P_{\bar{U}M}} \left[ \sum_{m=1}^M Q_{f_1}(m|v, \bar{U}^M)\|\bar{U}_m - v\|^2 \right] \tag{75}$$

$$= D(v, f_1, P_{\bar{U}M}). \tag{76}$$

for all $v$. This concludes the proof. $\qquad\square$

# D  Proof of Lemma 3.5

*Proof.* For $v, v' \in \mathbb{S}^{d-1}$, let $A_0$ be an orthonormal matrix such that $v' = A_0 v$. Let $f_2$ be a randomized encoder such that

$$f_2(v, U^M) = f(Av, AU^M) \tag{77}$$

for uniform random orthonormal matrix. Then,

$$Q_{f_2}(m|v, U^M) = \mathbb{E}_A \left[ Q_f(m|Av, AU^M) \right]. \tag{78}$$

Similar to the previous proofs, $Q_{f_2}$ is a well-defined probability distribution, and $f_2$ is unbiased as well as $\varepsilon$-LDP. Since $P_{UM}$ is rotationally symmetric and $f_2$ is also randomized via the uniform random orthogonal matrix, we have

$$D(v', f_2, P_{UM}) = D(A_0 v, f_2, P_{UM}) = D(v, f_2, P_{UM}). \tag{79}$$

Compared to a given randomizer $f$, we have

$$\mathrm{Err}(f, P_{UM}) \geq \mathbb{E}_A \left[ D(Av, f, P_{UM}) \right] \tag{80}$$

$$= \mathbb{E}_{A, P_{UM}} \left[ \sum_{m=1}^M Q_f(m|Av, U^M)\|Av - U^M\|^2 \right] \tag{81}$$

$$= \mathbb{E}_{A, P_{UM}} \left[ \sum_{m=1}^M Q_f(m|Av, U^M)\|v - A^\intercal U^M\|^2 \right] \tag{82}$$

$$= \mathbb{E}_{A, P_{UM}} \left[ \sum_{m=1}^M Q_f(m|Av, AU^M)\|v - U^M\|^2 \right] \tag{83}$$

$$= \mathbb{E}_{P_{UM}} \left[ \sum_{m=1}^M \mathbb{E}_A \left[ Q_f(m|Av, AU^M) \right] \|v - U^M\|^2 \right] \tag{84}$$

$$= D(v, f_2, P_{UM}) \tag{85}$$

for all $v \in \mathbb{S}^{d-1}$. This concludes the proof. $\qquad\square$

# E Proof of Theorem 3.6

*Proof.* The rotationally symmetric simplex codebook with normalization constant $r$ is $(rAs_1, \ldots, rAs_M)$. Let $f$ be the unbiased encoder satisfying $\varepsilon$-LDP. Let $Q_{\max} = \max Q_f(m|v, rAs^M)$ and $Q_{\min} = \min Q_f(m|v, rAs^M)$, our objective is to demonstrate that $Q_{\max}$ is less than or equal to $e^\varepsilon Q_{\min}$. We will employ a proof by contradiction to establish this. Suppose $Q_f(m_1|v_1, rA_1s^M) > e^\varepsilon Q_f(m_2|v_2, rA_2s^M)$ for some $m_1, v_1, A_1, m_2, v_2,$ and $A_2$. Let $\tilde{A}$ be the row switching matrix where $r\tilde{A}A_1s_{m_1} = rA_1s_{m_2}$ and $r\tilde{A}A_1s_{m_2} = rA_1s_{m_1}$, then we have

$$Q_f(m_1|v_1, rA_1s^M) = Q_f(m_2|\tilde{A}v_1, r\tilde{A}A_1s^M). \tag{86}$$

We further let $A'$ be an orthogonal matrix such that $A'\tilde{A}A_1 = A_2$, then

$$Q_f(m_2|\tilde{A}v_1, r\tilde{A}A_1s^M) = Q_f(m_2|A'\tilde{A}v_1, rA'\tilde{A}A_1s^M) \tag{87}$$

$$= Q_f(m_2|A'\tilde{A}v_1, rA_2s^M) \tag{88}$$

If we let $v_1' = A'\tilde{A}v_1$, then

$$Q_f(m_2|v_1', rA_2s^M) = Q_f(m_1|v_1, rA_1s^M) \tag{89}$$

$$> e^\varepsilon Q_f(m_2|v_2, rA_2s^M), \tag{90}$$

which contradicts the $\varepsilon$-LDP constraint.

For an unbiased encoder, the error is

$$\mathbb{E}_{P_{U^M}}\left[\sum_{m=1}^M \|U_m - v\|^2 Q_f(m|v, U^M)\right] = \mathbb{E}_{P_{U^M}}\left[\sum_{m=1}^M \|U_m\|^2 Q_f(m|v, U^M)\right] - 1 \tag{91}$$

$$= r^2 - 1. \tag{92}$$

Thus, we need to find $r$ that minimizes the error.

On the other hand, the encoder needs to satisfy unbiasedness. Without loss of generality, we assume $v = e_1$, then we need

$$\mathbb{E}_A\left[\sum_{m=1}^M rAs_m Q_f(m|e_1, rAs^M)\right] = e_1, \tag{93}$$

where the expectation is with respect to the random orthonormal matrix $A$. If we focus on the first index of the vector, then

$$r \times \mathbb{E}_a\left[\sum_{m=1}^M a^\mathsf{T}s_m Q_f(m|e_1, rAs^M)\right] = 1, \tag{94}$$

where $a^\mathsf{T} = (a_1, \ldots, a_d)$ is the first row of $A$ and has uniform distribution on the sphere $\mathbb{S}^{d-1}$. Thus, it is clear that assigning higher probability (close to $Q_{\max}$) to the larger $a^\mathsf{T}s_m$.

If $Q_{\max}$ is strictly smaller than $e^\varepsilon Q_{\min}$, then we can always scale up the larger probabilities and scale down the lower probabilities to keep the probability sum to one (while decreasing the error). Hence, we can assume that $Q_{\min} = q_0$ and $Q_{\max} = e^\varepsilon q_0$ for some $1 > q_0 > 0$.

Now, let $k$ be such that

$$(M - \lfloor k \rfloor - 1)q_0 + q_i + \lfloor k \rfloor e^\varepsilon q_0 = 1, \tag{95}$$

where $q_i$ is an intermediate value such that $q_i \in [q_0, e^\varepsilon q_0]$. Then, the optimal strategy is clear: (i) assign $e^\varepsilon q_0$ to $\lfloor k \rfloor$-th closest codewords $s_m$'s, (ii) assign $q_i$ to the $(\lfloor k \rfloor + 1)$-th closest codeword, and (iii) assign $q_0$ to the remaining codewords. This implies that the $k$-closest coding is optimal. □

# F Proof of Lemma 3.7

*Proof.* Following (28) with $U_m = As_m$ and $v = e_1$, we have

$$r_k \frac{e^\varepsilon - 1}{ke^\varepsilon + (M - k)} \mathbb{E}\left[\sum_{m \in T_k(e_1, A \cdot S)} A \cdot s_m\right]$$

$$= r_k \frac{e^\varepsilon - 1}{ke^\varepsilon + (M-k)} \mathbb{E} \left[ \sum_{m \in \arg\max_k(\{\langle e_1, As_1 \rangle, ..., \langle e_1, As_M \rangle\})} A \cdot s_m \right]$$

$$= e_1.$$

By focusing on the first coordinate of the above equation and observing that $\langle e_1, As_M \rangle = \langle a, s_m \rangle$ where $a$ is the first row of the rotation matrix $A$, we must have

$$r_k \cdot \frac{e^\varepsilon - 1}{ke^\varepsilon + (M-k)} \mathbb{E}_{a \sim \mathsf{unif}(\mathbf{S}^{d-1})} \left[ \sum_{m \in \mathsf{Top}_k(\{\langle a, s_1 \rangle, ..., \langle a, s_M \rangle\})} \langle a, s_m \rangle \right] = 1. \tag{96}$$

Note that since $A$ is a random orthogonal matrix drawn from the Haar measure on $SO(d)$, $a$ is distributed uniformly over the unit sphere $\mathbb{S}^{d-1}$.

Next, observe that by definition,

$$s_m = \frac{M}{\sqrt{M(M-1)}} e_m - \frac{1}{\sqrt{M(M-1)}} \mathbf{1}_M,$$

where $\mathbf{1}_M = (\underbrace{1, 1, ..., 1}_{M \text{ entries}}, 0, ..., 0) \in \{0, 1\}^d$ (that is, $(\mathbf{1}_M)_m = \mathbb{1}_{\{m \leq M\}}$). Therefore,

$$\langle a, s_m \rangle = \frac{M}{\sqrt{M(M-1)}} a_m - \frac{1}{\sqrt{M(M-1)}} \langle a, \mathbf{1}_M \rangle,$$

and hence plugging in (96) yields

$$r_k \cdot \frac{e^\varepsilon - 1}{ke^\varepsilon + (M-k)} \mathbb{E}_{a \sim \mathsf{unif}(\mathbf{S}^{d-1})} \left[ \sum_{m \in \mathsf{Top}_k(\{\langle a, s_1 \rangle, ..., \langle a, s_M \rangle\})} \langle a, s_m \rangle \right]$$

$$= r_k \cdot \frac{e^\varepsilon - 1}{ke^\varepsilon + (M-k)} \cdot \frac{M}{\sqrt{M(M-1)}} \mathbb{E}_{a \sim \mathsf{unif}(\mathbf{S}^{d-1})} \left[ \sum_{i=1}^{k} a_{(i|M)} - \frac{k}{M} \langle a, \mathbf{1}_M \rangle \right]$$

$$= r_k \cdot \frac{e^\varepsilon - 1}{ke^\varepsilon + (M-k)} \cdot \sqrt{\frac{M}{M-1}} \cdot \underbrace{\mathbb{E}_{a \sim \mathsf{unif}(\mathbf{S}^{d-1})} \left[ \sum_{i=1}^{k} a_{(i|M)} \right]}_{:=C_k},$$

where (1) $a_{(i|M)}$ denotes the $i$-th largest entry of the first $M$ coordinates of $a$ and (2) the last equality holds since $a$ is uniformly distributed over $\mathbb{S}^{d-1}$. $\qquad \square$

## G   Proof of Lemma 3.8

*Proof.* First of all, observe that

$$\mathbb{E}_{a \sim \mathsf{unif}(\mathbb{S}^{d-1})} \left[ \sum_{i=1}^{k} a_{(i|M)} \right]$$

$$= \mathbb{E}_{a \sim \mathsf{unif}(\mathbb{S}^{d-1})} \left[ \mathbb{E} \left[ \sum_{i=1}^{k} a_{(i|M)} \middle| \sum_{i=1}^{M} a_i^2 \right] \right]$$

$$\overset{(a)}{=} \mathbb{E}_{a \sim \mathsf{unif}(\mathbb{S}^{d-1})} \left[ \sqrt{\sum_{i=1}^{M} a_i^2} \cdot \mathbb{E}_{(a_1', ..., a_M') \sim \mathsf{unif}(\mathbb{S}^{M-1})} \left[ \sum_{i=1}^{k} a_{(i)}' \right] \right]$$

$$= \underbrace{\mathbb{E}_{a \sim \mathsf{unif}(\mathbb{S}^{d-1})} \left[ \sqrt{\sum_{i=1}^{M} a_i^2} \right]}_{(i)} \cdot \underbrace{\mathbb{E}_{(a_1', ..., a_M') \sim \mathsf{unif}(\mathbb{S}^{M-1})} \left[ \sum_{i=1}^{k} a_{(i)}' \right]}_{(ii)},$$

where (a) holds due to the spherical symmetry of $a$. Next, we bound (i) and (ii) separately.

**Claim G.1** (Bounding (i)). *For any $d \geq M > 2$, it holds that*

$$\sqrt{\frac{M-2}{d-2}} \leq \mathbb{E}_{a \sim \text{unif}(\mathbb{S}^{d-1})} \left[ \sqrt{\sum_{i=1}^{M} a_i^2} \right] \leq \sqrt{\frac{M}{d-2}}. \tag{97}$$

**Proof of Claim G.1.** Observe that when $a$ is distributed uniformly over $\mathbb{S}^{d-1}$, it holds that

$$(a_1, a_2, ..., a_d) \overset{d}{=} \left( \frac{Z_1}{\sqrt{\sum_{i=1}^{d} Z_i^2}}, \frac{Z_2}{\sqrt{\sum_{i=1}^{d} Z_i^2}}, ..., \frac{Z_d}{\sqrt{\sum_{i=1}^{d} Z_i^2}} \right),$$

where $A \overset{d}{=} B$ denotes $A$ and $B$ have the same distribution, and $Z_1, ..., Z_d \overset{\text{i.i.d.}}{\sim} \mathcal{N}(0,1)$. As a result, we must have

$$\mathbb{E}_{a \sim \text{unif}(\mathbb{S}^{d-1})} \left[ \sqrt{\sum_{i=1}^{M} a_i^2} \right] = \mathbb{E}_{Z_1,...,Z_M \overset{\text{i.i.d.}}{\sim} \mathcal{N}(0,1)} \left[ \sqrt{\frac{\sum_{i=1}^{M} Z_i^2}{\sum_{i=1}^{M} Z_i^2 + \sum_{i'=M+1}^{d} Z_{i'}^2}} \right].$$

By Jensen's inequality, it holds that

$$\mathbb{E}_{Z_1,...,Z_M \overset{\text{i.i.d.}}{\sim} \mathcal{N}(0,1)} \left[ \sqrt{\frac{\sum_{i=1}^{M} Z_i^2}{\sum_{i=1}^{M} Z_i^2 + \sum_{i'=M+1}^{d} Z_{i'}^2}} \right]$$

$$= \mathbb{E}_{Z_1,...,Z_M \overset{\text{i.i.d.}}{\sim} \mathcal{N}(0,1)} \left[ \sqrt{\frac{1}{1 + \frac{\sum_{i'=M+1}^{d} Z_{i'}^2}{\sum_{i=1}^{M} Z_i^2}}} \right]$$

$$\overset{(a)}{\geq} \sqrt{\frac{1}{1 + \mathbb{E}_{Z_1,...,Z_M \overset{\text{i.i.d.}}{\sim} \mathcal{N}(0,1)} \left[ \frac{\sum_{i'=M+1}^{d} Z_{i'}^2}{\sum_{i=1}^{M} Z_i^2} \right]}}$$

$$\overset{(b)}{=} \sqrt{\frac{1}{1 + \frac{d-M}{M-2}}}$$

$$= \sqrt{\frac{M-2}{d-2}},$$

where (a) holds since $x \mapsto \sqrt{1/(1+x)}$ is a convex mapping for $x > 0$, and (b) holds due to the fact that $\sum_i Z_i^2$ follows from a $\chi^2$ distribution and that the ratio of two independent $\chi^2$ random variables follows an $F$-distribution.

On the other hand, it also holds that

$$\mathbb{E}_{Z_1,...,Z_M \overset{\text{i.i.d.}}{\sim} \mathcal{N}(0,1)} \left[ \sqrt{\frac{\sum_{i=1}^{M} Z_i^2}{\sum_{i=1}^{M} Z_i^2 + \sum_{i'=M+1}^{d} Z_{i'}^2}} \right]$$

$$\overset{(a)}{\leq} \sqrt{\mathbb{E}_{Z_1,...,Z_M \overset{\text{i.i.d.}}{\sim} \mathcal{N}(0,1)} \left[ \frac{\sum_{i=1}^{M} Z_i^2}{\sum_{i=1}^{M} Z_i^2 + \sum_{i'=M+1}^{d} Z_{i'}^2} \right]}$$

$$= \sqrt{\mathbb{E}_{Z_1,...,Z_M \overset{\text{i.i.d.}}{\sim} \mathcal{N}(0,1)} \left[ 1 - \frac{\sum_{i=M+1}^{d} Z_{i'}^2}{\sum_{i=1}^{M} Z_i^2 + \sum_{i'=M+1}^{d} Z_{i'}^2} \right]}$$

$$= \sqrt{1 - \mathbb{E}_{Z_1,...,Z_M \overset{\text{i.i.d.}}{\sim} \mathcal{N}(0,1)} \left[ \frac{1}{1 + \frac{\sum_{i=1}^{M} Z_i^2}{\sum_{i=M+1}^{d} Z_{i'}^2}} \right]}$$

$$\overset{(b)}{\leq} \sqrt{1 - \frac{1}{1 + \mathbb{E}_{Z_1,...,Z_M \overset{\text{i.i.d.}}{\sim} \mathcal{N}(0,1)}\left[\frac{\sum_{i=1}^M Z_i^2}{\sum_{i=M+1}^d Z_{i'}^2}\right]}}$$

$$\overset{(c)}{=} \sqrt{1 - \frac{1}{1 + \frac{M}{d-M-2}}}$$

$$= \sqrt{\frac{M}{d-2}},$$

where (a) holds since $\sqrt{\cdot}$ is concave, (b) holds since $x \mapsto \frac{1}{1+x}$ is convex, and (c) again is due to the fact that the ratio of two independent $\chi^2$ random variables follows an $F$-distribution.

**Claim G.2** (Bounding (ii)). *As long as*

- $k \geq 400 \cdot \log 10$,

- $\log(M/k) \geq \left(\frac{10^3 \pi \log 2}{9}\right)^2$,

*it holds that*

$$\sqrt{\frac{k \log\left(\frac{M}{k}\right)}{24\pi \log 2M}} \leq \mathbb{E}_{(a'_1,...,a'_M) \sim \text{unif}(\mathbb{S}^{M-1})}\left[\sum_{i=1}^k a'_{(i)}\right] \leq \sqrt{\frac{4k \log M}{M}}. \tag{98}$$

**Proof of Claim G.2.** We start by re-writing $a'$:

$$(a'_1, a'_2, ..., a'_M) \overset{d}{=} \left(\frac{Z_1}{\sqrt{\sum_{i=1}^M Z_i}}, \frac{Z_2}{\sqrt{\sum_{i=1}^M Z_i}}, ...., \frac{Z_M}{\sqrt{\sum_{i=1}^M Z_i}}\right).$$

This yields that

$$(a'_{(1)}, a'_{(2)}, ..., a'_{(k)}) \overset{d}{=} \left(\frac{Z_{(1)}}{\sqrt{\sum_{i=1}^M Z_i^2}}, \frac{Z_{(2)}}{\sqrt{\sum_{i=1}^M Z_i^2}}, ...., \frac{Z_{(k)}}{\sqrt{\sum_{i=1}^M Z_i^2}}\right),$$

and hence

$$\mathbb{E}_{(a'_1,...,a'_M) \sim \text{unif}(\mathbb{S}^{M-1})}\left[\sum_{i=1}^k a'_{(i)}\right] = \mathbb{E}_{Z_1,...,Z_M \overset{\text{i.i.d.}}{\sim} \mathcal{N}(0,1)}\left[\frac{1}{\sqrt{\sum_{i=1}^M Z_i^2}}\sum_{i=1}^k Z_{(i)}\right].$$

**Upper bound.** To upper bound the above, observe that

$$\mathbb{E}_{Z_1,...,Z_M \overset{\text{i.i.d.}}{\sim} \mathcal{N}(0,1)}\left[\frac{1}{\sqrt{\sum_{i=1}^M Z_i^2}}\sum_{i=1}^k Z_{(i)}\right] \leq k\mathbb{E}_{Z_1,...,Z_M \overset{\text{i.i.d.}}{\sim} \mathcal{N}(0,1)}\left[\frac{1}{\sqrt{\sum_{i=1}^M Z_i^2}}Z_{(1)}\right].$$

Let $\mathcal{E}_1 := \left\{(Z_1, ..., Z_M)| \sum_{i=1}^M Z_i^2 \leq M(1-\gamma)\right\}$ where $\gamma > 0$ will be optimized later. Then it holds that

$$\Pr\{\mathcal{E}_1\} \leq e^{-\frac{M\gamma^2}{4}}. \tag{99}$$

On the other hand, the Borell-TIS inequality ensures

$$\Pr\left\{\left|Z_{(1)} - \mathbb{E}\left[Z_{(1)}\right]\right| > \xi\right\} \leq 2e^{-\frac{\xi^2}{2\sigma^2}}, \tag{100}$$

where $Z_i \sim \mathcal{N}(0, \sigma^2)$ (in our case, $\sigma = 1$). Since $\mathbb{E}\left[Z_{(1)}\right] \leq \sqrt{2 \log M}$, it holds that

$$\Pr\left\{Z_{(1)} \geq \sqrt{2 \log M} + \xi\right\} \leq 2e^{-\xi^2}.$$

Therefore, define $\mathcal{E}_2 := \left\{Z_{(1)} \geq \sqrt{2 \log M} + \xi\right\}$ and we obtain

$$\mathbb{E}_{Z_1,...,Z_M \overset{\text{i.i.d.}}{\sim} \mathcal{N}(0,1)}\left[\frac{1}{\sqrt{\sum_{i=1}^M Z_i^2}} \sum_{i=1}^k Z_{(i)}\right]$$

$$\leq k\mathbb{E}_{Z_1,...,Z_M \overset{\text{i.i.d.}}{\sim} \mathcal{N}(0,1)}\left[\frac{1}{\sqrt{\sum_{i=1}^M Z_i^2}} Z_{(1)}\right]$$

$$\leq k \cdot \left(\mathbb{E}\left[\frac{Z_{(1)}}{\sqrt{\sum_{i=1}^M Z_i^2}} \middle| \mathcal{E}_1 \cap \mathcal{E}_2\right] + \sup_{z_1,...,z_m}\left(\frac{z_{(1)}}{\sqrt{\sum_{i=1}^M z_i^2}}\right) \cdot \Pr\left(\mathcal{E}_1^c \cup \mathcal{E}_2^c\right)\right)$$

$$\leq k \cdot \left(\frac{\sqrt{2 \log M} + \xi}{M(1 - \gamma)} + 1 \cdot \left(e^{-M\gamma^2/4} + 2e^{-\xi^2}\right)\right)$$

$$\leq k \cdot \left(\frac{\sqrt{2 \log M} + \sqrt{\log(M)}}{0.9 \cdot M} + 1 \cdot \left(e^{-M/400} + 2/M\right)\right)$$

$$= \Theta\left(\frac{k\sqrt{\log M}}{M}\right),$$

where the last inequality holds by picking $\gamma = 0.1$ and $\xi = \sqrt{\log M}$.

**Lower bound.** The analysis of the lower bound is more sophisticated. To begin with, let

$$\mathcal{E}_M := \left\{(Z_1, ..., Z_M) \middle| \sum_{i=1}^M Z_i^2 \in [M(1 - \gamma), M(1 + \gamma)]\right\}$$

denote the good event such that the denominator of our target is well-controlled, where $\gamma > 0$ again will be optimized later. By the concentration of $\chi^2$ random variables, it holds that

$$\Pr\{\mathcal{E}_M^c\} \leq e^{-\frac{M}{2}(\gamma - \log(1+\gamma))} + e^{-\frac{M\gamma^2}{4}} \leq e^{-\frac{M}{2}\left(1 - \frac{1}{\sqrt{1+\gamma}}\right)\gamma} + e^{-\frac{M\gamma^2}{4}} \leq 2e^{-\frac{M\gamma^2}{4}}. \tag{101}$$

Next, to lower bound $\sum_{i=1}^k Z_{(i)}$, we partition $(Z_1, Z_2, ..., Z_M)$ into $k$ blocks $B_1, B_2, ..., B_k$ where each block contains at least $N = \lfloor M/k \rfloor$ samples: $B_j := [(j-1) \cdot N + 1 : j \cdot N]$ for $j \in [k-1]$ and $B_k = [M] \setminus \left(\bigcup_{j=1}^{k-1} B_j\right)$. Define $\tilde{Z}_{(1)}^{(j)}$ be the maximum samples in the $j$-th block: $\tilde{Z}_{(1)}^{(j)} := \max_{i \in B_j} Z_i$. Then, it is obvious that

$$\sum_{i=1}^k Z_{(i)} \geq \sum_{j=1}^k \tilde{Z}_{(1)}^{(j)}.$$

To this end, we define $\mathcal{E}_1$ to be the good event that 90% of $\tilde{Z}_{(1)}^{(j)}$'s are large enough (i.e., concentrated to the expectation):

$$\mathcal{E}_1 := \left\{\left|\left\{j \in [k] \middle| \tilde{Z}_{(1)}^{(j)} \geq \frac{\sqrt{\log N}}{\sqrt{\pi \log 2}} - \log 100\right\}\right| > 0.9k\right\}.$$

Note that by the Borell-TIS inequality, for any $j \in [k]$,

$$\Pr\left\{\tilde{Z}_{(1)}^{(j)} \geq \frac{\sqrt{\log N}}{\sqrt{\pi \log 2}} - \xi\right\} \geq 1 - 2e^{-\xi^2},$$

so setting $\xi = \log 100$ implies $\Pr\left\{\tilde{Z}_{(1)}^{(j)} \geq \frac{\sqrt{\log N}}{\sqrt{\pi \log 2}} - \xi\right\} \geq 0.98$. Since blocks are independent with each other, applying Hoeffding's bound yields

$$\Pr\{\mathcal{E}_1\} \geq 1 - \Pr\{\mathsf{Binom}(k, 0.98) \leq 0.9\} \geq 1 - e^{-k(0.08)^2} \geq 0.9,$$

when $k \geq 400 \cdot \log 10 \geq \log 10/0.08^2$.

Next, we define a "not-too-bad" event where $\sum_{j=1}^{k} \tilde{Z}_{(1)}^{(j)}$ is not catastrophically small:

$$\mathcal{E}_2 := \left\{\sum_{j=1}^{k} \tilde{Z}_{(1)}^{(j)} \geq -\frac{k}{\sqrt{M}}\xi\right\},$$

for some $\xi > 0$ to be optimized later. Observe that $\mathcal{E}_2$ holds with high probability:

$$\Pr\{\mathcal{E}_2\} \overset{(a)}{\geq} \Pr\left\{\frac{k}{M}\sum_{i=1}^{M} Z_i \geq -\frac{k}{\sqrt{M}}\xi\right\}$$

$$\overset{(b)}{\geq} 1 - e^{-\xi^2/2},$$

where (a) holds since the each of the top-$k$ values must be greater than $k$ times the average, and (b) holds due to the Hoeffding's bound on the sum of i.i.d. Gaussian variables.

Lastly, a trivial bound implies that

$$\inf_{a \in \mathbb{S}^{M-1}} \sum_{i=1}^{k} a_{(i)} \geq -\frac{k}{\sqrt{M}}.$$

Now, we are ready to bound $\mathbb{E}_{Z_1,\dots,Z_M \overset{\text{i.i.d.}}{\sim} \mathcal{N}(0,1)}\left[\frac{1}{\sqrt{\sum_{i=1}^{M} Z_i^2}}\sum_{i=1}^{k} Z_{(i)}\right]$. We begin by decomposing it into three parts:

$$\mathbb{E}_{Z_1,\dots,Z_M \overset{\text{i.i.d.}}{\sim} \mathcal{N}(0,1)}\left[\frac{\sum_{i=1}^{k} Z_{(i)}}{\sqrt{\sum_{i=1}^{M} Z_i^2}}\right] = \Pr\{\mathcal{E}_1 \cap \mathcal{E}_2 \cap \mathcal{E}_M\} \cdot \mathbb{E}\left[\frac{\sum_{i=1}^{k} Z_{(i)}}{\sqrt{\sum_{i=1}^{M} Z_i^2}} \,\middle|\, \mathcal{E}_1 \cap \mathcal{E}_2 \cap \mathcal{E}_M\right]$$

$$+ \Pr\{\mathcal{E}_1^c \cap \mathcal{E}_2 \cap \mathcal{E}_M\} \cdot \mathbb{E}\left[\frac{\sum_{i=1}^{k} Z_{(i)}}{\sqrt{\sum_{i=1}^{M} Z_i^2}} \,\middle|\, \mathcal{E}_1^c \cap \mathcal{E}_2 \cap \mathcal{E}_M\right]$$

$$+ \Pr\{\mathcal{E}_2^c \cup \mathcal{E}_M^c\} \cdot \mathbb{E}\left[\frac{\sum_{i=1}^{k} Z_{(i)}}{\sqrt{\sum_{i=1}^{M} Z_i^2}} \,\middle|\, \mathcal{E}_2^c \cup \mathcal{E}_M^c\right].$$

We bound these three terms separately. To bound the first one, observe that condition on $\mathcal{E}_1 \cap \mathcal{E}_2$, $\sum_{i=1}^{k} Z_{(i)} \geq \tilde{Z}_{(1)}^{(j)} \geq 0.9k\sqrt{\frac{\log N}{\pi \log 2}} - \frac{k}{\sqrt{M}}\gamma$. As a result,

$$\Pr\{\mathcal{E}_1 \cap \mathcal{E}_2 \cap \mathcal{E}_M\} \cdot \mathbb{E}\left[\frac{\sum_{i=1}^{k} Z_{(i)}}{\sqrt{\sum_{i=1}^{M} Z_i^2}} \,\middle|\, \mathcal{E}_1 \cap \mathcal{E}_2 \cap \mathcal{E}_M\right]$$

$$\geq \frac{0.9k\sqrt{\frac{\log N}{\pi \log 2}} - \frac{k}{\sqrt{M}}\gamma}{\sqrt{M(1+\gamma)}} \cdot \left(1 - \left(0.1 + e^{-\xi^2/2} + 2e^{-M\gamma^2/4}\right)\right). \tag{102}$$

To bound the second term, observe that under $\mathcal{E}_2$,

$$\sum_{i=1}^{k} Z_{(i)} \geq -\frac{k}{\sqrt{M}}\xi,$$

so we have

$$\Pr\left\{\mathcal{E}_2 \cap \mathcal{E}_1^c \cap \mathcal{E}_M\right\} \cdot \mathbb{E}\left[\frac{\sum_{i=1}^k Z_{(i)}}{\sqrt{\sum_{i=1}^M Z_i^2}}\middle| \mathcal{E}_2 \cap \mathcal{E}_1^c \cap \mathcal{E}_M\right]$$

$$\geq \Pr\left\{\mathcal{E}_2 \cap \mathcal{E}_1^c \cap \mathcal{E}_M\right\} \cdot \left(-\frac{k}{\sqrt{M^2(1-\gamma)}}\xi\right)$$

$$\geq \Pr\left\{\mathcal{E}_1^c\right\} \cdot \left(-\frac{k}{\sqrt{M^2(1-\gamma)}}\xi\right)$$

$$\geq 0.1 \cdot \left(-\frac{\xi\sqrt{k}}{\sqrt{M^2(1-\gamma)}}\right). \tag{103}$$

For the third term, it holds that

$$\Pr\left\{\mathcal{E}_2^c \cup \mathcal{E}_M^c\right\} \cdot \mathbb{E}\left[\frac{\sum_{i=1}^k Z_{(i)}}{\sqrt{\sum_{i=1}^M Z_i^2}}\middle| \mathcal{E}_2^c \cup \mathcal{E}_M^c\right]$$

$$\geq \Pr\left\{\mathcal{E}_2^c \cup \mathcal{E}_M^c\right\} \cdot \inf_{a \in \mathbb{S}^{M-1}} \sum_{i=1}^k a_{(i)}$$

$$\geq -\Pr\left\{\mathcal{E}_2^c \cup \mathcal{E}_M^c\right\} \cdot \frac{k}{\sqrt{M}}$$

$$\geq -\left(e^{-\xi^2/2} + e^{-M\gamma^2/4}\right) \cdot \frac{k}{\sqrt{M}} \tag{104}$$

Combining (102), (103), and (104) together, we arrive at

$$\mathbb{E}_{Z_1,\ldots,Z_M \overset{\text{i.i.d.}}{\sim} \mathcal{N}(0,1)}\left[\frac{\sum_{i=1}^k Z_{(i)}}{\sqrt{\sum_{i=1}^M Z_i^2}}\right]$$

$$\geq \frac{0.9k\left(\sqrt{\frac{\log N}{\pi \log 2}}\right) - \frac{k}{\sqrt{M}}\gamma}{\sqrt{M(1+\gamma)}} \cdot \left(1 - \left(0.1 + e^{-\xi^2/2} + 2e^{-M\gamma^2/4}\right)\right) - 0.1 \cdot \left(\frac{\xi\sqrt{k}}{\sqrt{M^2(1-\gamma)}}\right)$$

$$- \left(e^{-\xi^2/2} + e^{-M\gamma^2/4}\right) \cdot \frac{k}{\sqrt{M}}.$$

Finally, setting $\gamma = O\left(\frac{1}{\sqrt{M}}\right)$ and $\xi = O(1)$ yields the desired lower bound

$$C_{d,M,k} = \Omega\left(\frac{k\log N}{\sqrt{M}}\right).$$

$\square$

# H Additional Experimental Results

In Figure 2, we provide additional empirical results by sweeping the number of users $n$ from $2,000$ to $10,000$ on the left and sweeping the dimension $d$ from $200$ to $1,000$ on the right.

# I Additional Details on Prior LDP Schemes

For completeness, we provide additional details on prior LDP mean estimation schemes in this section, including `PrivUnit` [4], `SQKR` [6], `FT21` [12], and `MMRC` [30]. We skip prior work analyzing compression-privacy-utility tradeoffs that do not specifically focus on the distributed mean estimation problem [19, 20] or others that study frequency estimation [6, 11, 30].

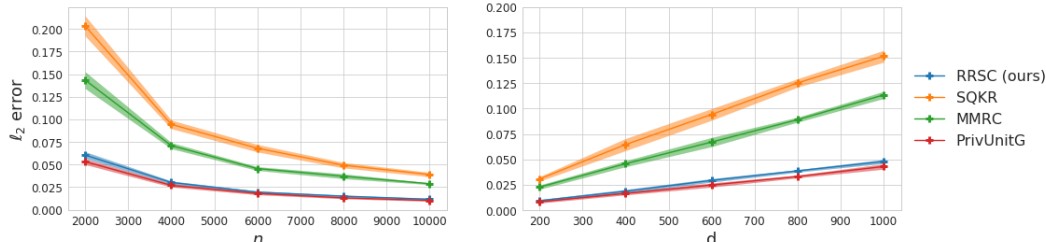

Figure 2: Comparison of `RRSC` with `SQKR` [6], `MMRC` [30], and `PrivUnitG` [2]. **(left)** $\ell_2$ error vs number of users $n$ with $d = 500$, $\varepsilon = 6$, and the number of bits is $b = \varepsilon = 6$. $k = 1$ for each $n$. **(right)** $\ell_2$ error vs dimension $d$ for $n = 5000$, $\varepsilon = 6$, and the number of bits is $b = \varepsilon = 6$. $k = 1$ for for each $d$.

## I.1  `PrivUnit` [4]

[2] considered the mean estimation problem under DP constraint (without communication constraint) when $\mathcal{X} = \mathbb{S}^{d-1} = \{v \in \mathbb{R}^d : \|v\|_1 = 1\}$. Since there is no communication constraint, they assumed canonical protocol where the random encoder is $f : \mathbb{S}^{d-1} \to \mathbb{R}^d$ and the decoder is a simple additive aggregator

$$g_n(f(v_1), \ldots, f(v_n)) = \frac{1}{n} \sum_{i=1}^n f(v_i).$$

The authors showed that PrivUnit is an exact optimal among the family of unbiased locally private procedures.

Recall that given an input vector $v \in \mathbb{S}^{d-1}$, the local randomized PrivUnit$(p, q)$ has the following distribution up to normalization:

$$\mathrm{PrivUnit}(p, q) \sim \begin{cases} Z | \langle Z, v \rangle \geq \gamma & \text{w.p. } p \\ Z | \langle Z, v \rangle < \gamma & \text{w.p. } 1 - p \end{cases}$$

where $Z$ has a uniform distribution on $\mathbb{S}^{d-1}$. Let $S_\gamma$ be the surface area of hypersphere cap $\{z \in \mathbb{S}^{d-1} | \langle z, v \rangle \geq \gamma\}$, with $S_{-1}$ representing the surface area of the $d$ dimensional hypersphere. We denoted $q = \Pr[Z_1 \leq \gamma] = (S_{-1} - S_\gamma)/S_{-1}$ (convention from [4, 2]). The normalization factor is required to obtain unbiasedness.

[2] also introduced `PrivUnitG`, which is a Gaussian approximation of `PrivUnit`. In this approach, $Z$ is sampled from an i.i.d. $\mathcal{N}(0, 1/d)$ distribution. This simplifies the process of determining more accurate parameters $p, q$, and $\gamma$. Consequently, in practical applications, `PrivUnitG` surpasses `PrivUnit` in performance owing to superior parameter optimization.

## I.2  `SQKR` [6]

Next, we outline the encoder and decoder of SQKR in this section. The encoding function mainly consists of three steps: (1) computing Kashin's representation, (2) quantization, and (3) sampling and privatization.

**Compute Kashin's representation**   A tight frame is a set of vectors $\{u_j\}_{j=1}^N \in \mathbb{R}^d$ that satisfy Parseval's identity, i.e. $\|v\|_2^2 = \sum_{j=1}^N \langle u_j, v \rangle^2$ for all $v \in \mathbb{R}^d$. We say that the expansion $v = \sum_{j=1}^N a_j u_j$ is a Kashin's representation of $x$ at level $K$ if $\max_j |a_j| \leq \frac{K}{\sqrt{N}} \|v\|_2$ [23]. [27] shows that if $N > (1 + \mu) d$ for some $\mu > 0$, then there exists a tight frame $\{u_j\}_{j=1}^N$ such that for any $x \in \mathbb{R}^d$, one can find a Kashin's representation at level $K = \Theta(1)$. This implies that we can represent the local vector $v$ with coefficients $\{a_j\}_{j=1}^N \in [-c/\sqrt{d}, c/\sqrt{d}]^N$ for some constants $c$ and $N = \Theta(d)$.

**Quantization**  In the quantization step, each client quantizes each $a_j$ into a 1-bit message $q_j \in \left\{-c/\sqrt{d}, c/\sqrt{d}\right\}$ with $\mathbb{E}[q_j] = a_j$. This yields an unbiased estimator of $\{a_j\}_{j=1}^N$, which can be described in $N = \Theta(d)$ bits. Moreover, due to the small range of each $a_j$, the variance of $q_j$ is bounded by $O(1/d)$.

**Sampling and privatization**  To further reduce $\{q_j\}$ to $k = \min(\lceil\varepsilon\rceil, b)$ bits, client $i$ draws $k$ independent samples from $\{q_j\}_{j=1}^N$ with the help of shared randomness, and privatizes its $k$ bits message via $2^k$-RR mechanism[36], yielding the final privatized report of $k$ bits, which it sends to the server.

Upon receiving the report from client $i$, the server can construct unbiased estimators $\hat{a}_j$ for each $\{a_j\}_{j=1}^N$, and hence reconstruct $\hat{v} = \sum_{j=1}^N \hat{a}_j u_j$, which yields an unbiased estimator of $v$. In [6], it is shown that the variance of $\hat{v}$ can be controlled by $O\left(d/\min\left(\varepsilon^2, \varepsilon, b\right)\right)$.

### I.3  `FT21` [12] and `MMRC` [30]

Both `FT21` and `MMRC` aim to simulate a given $\varepsilon$-LDP scheme. More concretely, consider an $\varepsilon$-LDP mechanism $q(\cdot|v)$ that we wish to compress, which in our case, `PrivUnit`. A number of candidates $u_1, \cdots, u_N$ are drawn from a fixed reference distribution $p(u)$ (known to both the client and the server), which in our case, uniform distribution on the sphere $\mathbb{S}^{d-1}$. Under `FT21` [12], these candidates are generated from an (exponentially strong) PRG, with seed length $\ell = \mathsf{polylog}(d)$. The client then performs rejection sampling and sends the seed of the sampled candidates to the server. See Algorithm 2 for an illustration.

---

**Algorithm 2** Simulating LDP mechanisms via rejection sampling [12]

---

**Inputs:** $\varepsilon$-LDP mechanism $q(\cdot|v)$, ref. distribution $p(\cdot)$, seeded PRG $G : \{0,1\}^\ell \to \{0,1\}^t$, failure probability $\gamma > 0$.

   $J = e^\varepsilon \ln(1/\gamma)$.
  **for** $j \in \{1, \cdots, J\}$ **do**
     Sample a random seed $s \in \{0,1\}^\ell$.
     Draw $u \leftarrow p(\cdot)$ using the PRG $G$ and the random seed $s$.
     Sample $b$ from Bernoulli $\left(\frac{q(u|v)}{e^\varepsilon \cdot p(u)}\right)$.
     **if** $b = 1$ **then**
       BREAK
     **end if**
  **end for**
**Output:** $s$

---

On the other hand, under `MRC` [30] the LDP mechanism is simulated via a minimal random coding technique [15]. Specifically, the candidates are generated via shared randomness, and the client performs an importance sampling and sends the index of the sampled one to the server, as illustrated in Algorithm 3. It can be shown that when the target mechanism is $\varepsilon$-LDP, the communication costs of both strategies are $\Theta(\varepsilon)$ bits. It is also worth noting that both strategies will incur some bias (though the bias can be made exponentially small as one increases the communication cost), and [30] provides a way to correct the bias when the target mechanism is `PrivUnit` (or general cap-based mechanisms).

**Algorithm 3** Simulating LDP mechanisms via importance sampling [30]

---

**Inputs:** $\varepsilon$-LDP mechanism $q(\cdot|v)$, ref. distribution $p(\cdot)$, # of candidates $M$

    Draw samples $u_1, \cdots, u_M$ from $p(u)$ using the shared source of randomness.

    **for** $k \in \{1, \cdots, M\}$ **do**

       $w(k) \leftarrow q(u_k|v)/p(u_k)$.

    **end for**

    $\pi_{\mathtt{MRC}}(\cdot) \leftarrow w(\cdot)/\sum_k w(k)$.

    Draw $k^* \leftarrow \pi_{\mathtt{MRC}}$.

**Output:** $k^*$

---

