# OpenReview forum: "Exact Optimality of Communication-Privacy-Utility Tradeoffs in Distributed Mean Estimation"
_NeurIPS.cc/2023/Conference — NeurIPS 2023 poster_

### Official Review · Reviewer_JsmV · 2023-06-29

**Soundness:** 4 excellent
**Presentation:** 3 good
**Contribution:** 3 good
**Rating:** 7
**Confidence:** 3

**Summary:**

The authors consider the problem of high dimensional mean estimation with communication and privacy constraints. For federated learning or distributed SGD, model updates must be communicated to central server, but as models become much larger this can be a bottleneck within the computation. As a result, previous works has considered the setting in which there is a restriction on the number of bits that can be communicated, which the authors also follow. Furthermore, the authors consider the differentially private setting adding another constraint. This setting was also considered in previous work, some of which achieved optimality up to constant factors. The authors improve upon that work and achieve exact optimality and this theoretical improvement is backed by their empirical experiments.

**Strengths:**

Improves upon previous work for a reasonably well-studied problem and achieve optimal tradeoff between communication-privacy-utility and further show how the previous work are special cases in their method.

**Weaknesses:**

Minor gripe that some of the notation could have been expanded upon more clearly (for example: P_U-almost) but can understand that page limits can add difficulty.

**Questions:**

None

**Limitations:**

The authors adequately addressed the limitations

---

> ### Author Rebuttal · Authors · 2023-08-09
>
> We thank the reviewer for carefully reading our manuscript and for providing positive and constructive feedback. We are glad that the reviewer acknowledged the improvements we provided over the prior order-optimal solutions and the importance of unifying the previous schemes under our exact-optimal solution. We also agree with the reviewer that some notations might need further clarification for some readers. We will provide more notational comments, e.g., $p$-almost means that it happens with probability 1 under measure $p$. We thank the reviewer for pointing out this.

---

> > ### Comment · Reviewer_JsmV · 2023-08-14
> >
> > Thanks for clarifying some of the notation and adding more detail in future versions of your work! The theorem statements in another rebuttal also added further clarity for me, and I appreciate the authors adding these details to future versions.

---

> > > ### Author Response · Authors · 2023-08-21
> > > **Thank you for your response to our rebuttal.**
> > >
> > > We thank the reviewer for their time reading our rebuttal. We will include the suggested details in the final manuscript.

---

### Official Review · Reviewer_jPit · 2023-07-06

**Soundness:** 3 good
**Presentation:** 3 good
**Contribution:** 3 good
**Rating:** 6
**Confidence:** 4

**Summary:**

The paper studies the distributed mean estimation (DME) problem with communication & privacy constraints.  The goal is to construct an unbiased estimate of a unit vector $v$ using $b$-bits that minimizes the mean squared error and provides $\epsilon$-LDP. It is well known that any scheme achieving the above communication will have to quantize the unit sphere using $M=2^b$ points. Further, to achieve $\epsilon$-LDP, a particular quantization point is chosen and returned according to an appropriate distribution.

The main contribution of this work is characterizing DME schemes that achieve optimal error in the presence of a communication budget and privacy constraints. The authors show that a random set of points generated using a rotationally symmetric distribution will achieve the optimal error. The intuition is that such a point set will be maximally separated and will most efficiently cover the sphere. $\epsilon$-LDP is achieved using a randomized response mechanism.

**Strengths:**

- Presents a DME scheme under exact communication constraints instead of asymptotic error bounds that are optimal.
- The idea of treating the k-nearest codewords equally instead of just the closest seems to provide improvements over the prior works. This clean trick can be of independent interest.

**Weaknesses:**

- The work provides sufficient conditions for a DME scheme to be optimal, i.e., an optimal scheme has a particular canonical setup.
These conditions are not necessary, and not all schemes satisfying the canonical setup achieve optimal error.

- Either some proofs have (probably fixable) errors, or I have misunderstood. So at least rephrasing or an elaborate explanation is required. Details are provided in the next section (Questions)

**Questions:**

1) As mentioned in the paper, the prior works of SQKR, MMRC, and FT21 can be seen as specific instantiations of random codebooks. Could it be possible that their schemes are exact-optimal as well and satisfy the conditions mentioned in this work? While a discussion is provided, can you comment on what condition is not met by each of these that confirms that they are not exact-optimal?

2) Line 217-220 leading to Eq 29 needs further justification. Intuitively, this holds if the k points are equidistant, but I am not sure why the k-nearest neighbors will give an unbiased estimate. In the current manuscript, this is just a statement and not a formal proof of the fact that "k-closest encoding consistently yields an unbiased scheme for any rotationally symmetric codebook." Further, it would be good to mention that for a rotationally symmetric codebook, the $\sum_m U_m = 0$.

3) Proof of Lemma 3.7 - What is top-k set? I am assuming it is the set of k-nearest neighbors to random a in s^M
Should the summation have only $s_m$ and not $a^Ts_m $?
Typo in eq 94 - the second summation should be over s_i.
Eq 96 shows that you get an unbiased estimate of $e_1$ and not $a$. Also, other coordinates cannot be ignored.
$r_k$ value is roughly $M$, so the error of the RRSC scheme will be $\sim M^2$. So for $M = d$, there is an error of $d^2$ which is higher than the order optimal ones $(O(d/\log d))$ for the same amount of allowed communication. A brief clarification on this would be great.

4) For the shared randomness setup, will the server/nodes have to regenerate the codebook for each invocation of the algorithm?

5) Some suggestions to improve the readability in my opinion would be:

- Fix errors in proofs or provide better justifications
- In Equation 26 the probability of choosing the k+1-th closest codeword is zero if k is an integer. This seems to be inconsistent with Algorithm 1, and the analysis that follows in Section 3.4
- Clarify what the `=' in Def 3.3 means. It can be confusing to think that the two codebooks are equal (permutations of each other) rather than their distributions being identical.

**Limitations:**

Yes

---

> ### Author Rebuttal · Authors · 2023-08-09
>
> We thank the reviewer for carefully reading our manuscript and providing positive and constructive feedback. The reviewer appreciated going beyond the asymptotic error bounds and found our k-nearest algorithm worthy of independent interest. The reviewer also asked clarifying questions, which we try to address below.
> ### **Exact optimality conditions**
> The reviewer is right that we show a sufficient condition as follows
> "There exists an exact-optimal canonical scheme with rotationally symmetric codebook." We will revisit our abstract and discussion to reflect this more accurately.
> ### **SQKR,FT21,MMRC**
> While it is clear from our experimental results that SQKR and MMRC fail to achieve exact optimality as RRSC significantly outperforms them, we now provide some intuitive reasoning why this is the case.
>
> **SQKR** produces a subset of indices $\mathcal{I}$ from $\lbrace1,2, \dots, d \rbrace$ with a size of $|\mathcal{I}|=k$
> utilizing the shared randomness. Then, the codebook defined as  $\lbrace\sum_{i\in \mathcal{I}} a_i u_i: a_i=c/\sqrt{d} \mbox{ or } -c/\sqrt{d} \rbrace$ represents a tight frame for Kashin's representation. It is essential to recognize that the distribution generating the codebook is not rotationally symmetric; it is only permutation symmetric.
> Moreover, the codewords are not maximally separated. Thus, we strongly believe that the reason why SQKR does not reach exact optimality is that it does not generate maximally separated codewords.
>
> **FT21** aims to simulate the continuous PrivUnit scheme via pseudo-random generator (PRG) and rejection sampling. As FT21 operates without employing shared randomness, it requires a minimum communication cost of $\Omega\left(\log d\right)$ bits to attain an unbiased estimator. In addition, as we noted in the paper, the codebook associated with FT21 depends on the PRG it uses. Although the specific PRGs in FT21 are not explicitly outlined, we believe our simplex code would outperform codebooks that correspond to standard PRGs. For high communication budget (large b), codewords based on PRGs may offer computational efficiency, but our focus in this paper centers on scenarios with smaller or moderate values of b, and hence our approach can yield a better (non-asymptotical) MSE at a manageable computational cost.
>
> **MMRC** introduces an importance sampling approach where samples are drawn based on a uniform distribution across the sphere,
> supplemented with a truncation technique. This is equivalent to producing i.i.d. codewords, which does not necessarily ensure maximal separation. We strongly believe that maximal separation of codewords, and hence the most effective coverage of the sphere, is important for exact optimality.
> ### **Eq (29)**
> We apologize for the confusion around lines 217-220. We try to clarify the step leading to Eq (29) here and will also revise it in the final manuscript. We note that in Eq (28), $\mathbb{E}[\sum_{m\in T_k}U_m]$ can only have $e_1$ as the remaining component since we pick the k-closest codewords $U_m$ (i.e., $m\in T_k$) and all directions other than $e_1$ are canceled out due to the rotational symmetry of the codewords. We hope this explanation clarifies the transition from Eq (28) to Eq (29). As suggested by the reviewer, we will clearly state in the final manuscript that $\mathbb{E}[\sum_m U_m]=0$ for all rotationally symmetric codebooks.
> ### **Proof of Lemma 3.7**
> We apologize for the confusion due to some typos. We will update the proof as follows. From (28), we want to find $r_k$ that satisfies $r_k \times \mathbb{E}[\frac{e^\epsilon-1}{ke^\epsilon+M-k}\sum_{m\in T_k(e_1,U^M)}U_m]=e_1$, where $U_m=As_m$. The condition $m\in T_k(e_1,U^M)$ implies that $<e_1,As_m>$ is one of the k-largest. Let $a_1^\intercal$ be the first row of the matrix A. Since $s_m$ is a vertex of the simplex, $<e_1,As_m>=a_1^\intercal s_m=\sum_{i=1}^M a_{1,i} s_{m,i}=a_{1,m} \sqrt{\frac{M}{M-1}}-\sum_{i=1}^M\sqrt{\frac{1}{M(M-1)}}a_{1,i}$. Since $\sum_{i=1}^M \sqrt{\frac{1}{M(M-1)}}a_{1,i}$ is not a function of m, the condition $m \in T_k(e_1,U^M)$ is equivalent to "$a_{1,m}$ is top-k among $\lbrace a_{1,i}\rbrace_{i=1}^M$". Also, due to the rotational symmetry, $\mathbb{E}[\sum_{i=1}^M \sqrt{\frac{1}{M(M-1)}}a_{1,i}]=0$. Thus, $r_k$ should satisfy $r_k \times \mathbb{E}[\frac{e^\epsilon-1}{ke^\epsilon+M-k} \sum_{m\in top-k}a_{1,m}]=1$, where $m\in$ top-k means that "$a_{1,m}$ is top-k among $\lbrace a_{1,i}\rbrace_{i=1}^M$".
> ### **Order of $r_k$**
> Since the role of k is similar to that of the threshold in PrivUnit, the optimal choice of k grows with O(M). Please also see Figure 1 in the attached pdf, which demonstrates how the optimal k (found by Algorithm 1 in the attached pdf) changes with M. On the other hand, $C_k$ is an expected sum of top-k coordinates of the unit vector, which is roughly $O(k \sqrt{1/d})$ for $k=O(M)$. Thus, unlike the reviewer's suspicion, $r_k$ **does not** scale with M; instead, it scales with $\sqrt{d}$. We will emphasize this better in the final version.
> ### **Non-Integer k**
> We note that in Eq (26), if k is an integer, the probability of selecting k+1-th closest codeword is simply $1/(ke^\varepsilon+M-k)$, which corresponds to the probability of the case "otherwise". The reason why we allowed k to be non-integer as well in Section 3.3 is to simplify the proof. In practice, however, both in Algorithm1 and in the experiments, we always pick an integer k. We also note that RRSC achieves impressive performance that matches PrivUnit even with an integer k.
> ### **Clarification on the Notation**
> The reviewer is indeed right that $\stackrel{(d)}{=}$ implies that the distributions are identical, which we will clearly state in the final manuscript.
>
> ---
> We again thank the reviewer for carefully reading our manuscript and for providing positive feedback. We hope we have addressed all the questions. We are happy to engage in further discussion with the reviewer to resolve any remaining concerns.

---

> > ### Comment · Reviewer_jPit · 2023-08-15
> >
> > Thanks for the detailed response.
> >
> > MMRC:
> > "MMRC introduces an importance sampling approach where samples are drawn based on a uniform distribution across the sphere, supplemented with a truncation technique. This is equivalent to producing i.i.d. codewords, which does not necessarily ensure maximal separation. We strongly believe that maximal separation of codewords, and hence the most effective coverage of the sphere, is important for exact optimality."
> >
> > iid codewords sampled from unit sphere = normalized high-dimensional Gaussians are with high probability good covers for the unit sphere.
> >
> > Eq 29:
> > For Eq29, my concern remains. It is still not clear to me without proof why the sum of the top $k$ closest vectors to $e_1$ in a rotationally symmetric codebook will cancel out in all other directions. Even if trivial, please include it for people like me.
> > For instance, in 2-dimensions, if I choose the codebook to be ${(0,1), (0,-1), (1,0), (-1,0)}$, and for $k=2$, the top $k$ closest to $e_1 = (1,0)$ will consist of ${(1,0), (0,1)}$ (or $(0,-1)$ depending on how you break ties.) which do not cancel out. For $k=3$, you will get this property.
> >
> > So I am guessing you mean to say that there exists a $k$, where the top-k will work. But even this will need proof and a bound on the value of $k$. Moreover, as stated in Algorithm 1, $k$ cannot be a part of the input.
> >
> > Order of $r_k$: I apologize for being technically challenged, but I could not find any attached pdf. However, I understand that the scaling of $r_k$ is roughly $(k + M)/C_k = O(M/C_k)$ ignoring the $\epsilon$ terms. While for large $k$, i.e., $k=O(M)$, $r_k$ scales as $\sqrt{d}$, for small $k =O(1)$, the scaling of $r_k$ seems to be roughly $O(M \sqrt{d})$.
> >
> > Non-integer $k$:  Equation 26 will translate to choosing the top $floor{k}+1$, and the probabilities will not all sum to 1. There is a tiny typo here that needs to be fixed.

---

> > > ### Author Response · Authors · 2023-08-15
> > > **Thank you for your response to our rebuttal.**
> > >
> > > We thank the reviewer for reading our rebuttal in detail and following up with further questions. We respond to each question below. The **pdf** with a new algorithm (to find the optimal k) and a new figure (that shows how k changes with increasing M) is **attached to our general response titled "Author Rebuttal by Authors" in the top -- before the reviewers' comments.**
> > >
> > > -----
> > > ##### **MMRC: "iid codewords sampled from unit sphere = normalized high-dimensional Gaussians are with high probability good covers for the unit sphere."**
> > > This holds true when sampling a large number of Gaussian vectors. However, when sampling a relatively small number of these vectors (as is our interest with a small $M$), it is essential to select the vectors judiciously to effectively cover the sphere.
> > >
> > > ---
> > > ##### **Eq 29:**
> > > The codebook is derived from a rotationally symmetric distribution through shared randomness,
> > > and any expectations are taken with respect to this distribution. Consider the codebook $\lbrace e_1, e_2, -e_1, -e_2\rbrace$ where $e_1 = (1,0)$ and $e_2 = (0, 1)$. This codebook is not rotationally symmetric. However, if $A$ is a uniformly sampled $2\times 2$ random rotation matrix, the codebook $\lbrace A e_1, A e_2, -A e_1, -A e_2\rbrace$ is rotationally symmetric. Given that $Ae_1$ has a uniform distribution on the sphere, the conditional expectation $\mathbb{E}[A e_1 |\mbox{$Ae_1$ is the closest to $e_1$}]$ can only contain the $e_1$ component. This is because the $e_2$ components cancel out due to symmetry.
> > >
> > > The reviewer is right that $k$ is not part of the input. As we show in Algorithm 1 in the attached pdf in our general response, the optimal $k$ is determined by $\varepsilon$, $d$, and $M$ by minimizing $r_k$.
> > >
> > >
> > > ----
> > > #### **Order of $r_k$:**
> > > For every $M$, we determine the best value of $k$ that yields the smallest $r_k$. We contend that the optimal selection of $k$ increases linearly with $M$, as evidenced in the provided PDF in general response. This leads to $r_k = O(\sqrt{d})$.
> > >
> > > ---
> > > #### **Non-integer $k$:**
> > > We apologize for the confusing notation. For top-$\lfloor k\rfloor$ candidates, we assign probability $e^\epsilon / (ke^\epsilon+M-k)$.
> > > For the ($\lfloor k\rfloor +1$)-th candidate, we assign probability $\frac{(k-\lfloor k\rfloor)(e^\epsilon-1)+1}{ke^\epsilon+M-k}$.
> > > For the rest $M-\lfloor k \rfloor -1$ candidates, we assign probability $1 / (ke^\epsilon+M-k)$.
> > > The sum of the probabilities is 1.
> > > \begin{align*}
> > >     \lfloor k \rfloor \times \frac{e^\epsilon}{ke^\epsilon+M-k}
> > >     + \frac{(k-\lfloor k\rfloor)(e^\epsilon-1)+1}{ke^\epsilon+M-k}
> > >     + (M-\lfloor k\rfloor -1) \times\frac{1}{ke^\epsilon+M-k} = 1
> > > \end{align*}
> > >
> > > ---
> > > We thank the reviewer for their time in reading the manuscript and the rebuttal in detail. We hope our response addresses the reviewer's points. If there is any other concern or confusion remaining, we are more than happy to discuss them. If our response is satisfactory for the reviewer, we kindly ask them to consider revisiting their score.

---

> > > > ### Comment · Reviewer_jPit · 2023-08-20
> > > >
> > > > Thanks for the clarification. I have a much better understanding now.
> > > >
> > > > Stupid question: Why cannot you set k=M/2? Put a larger mass on all the codewords c with c_1 > 0 (assuming you are quantizing e_1). Would you not get $r_k = O(\sqrt{d})$ with this?
> > > >
> > > > This can be generalized:
> > > > Instead of searching for the closest k = O(M) points, one could take O(1) random (or well-designed) hyperplanes and assign a higher probability to all the codewords with the same sign as the input. Keep adding hyperplanes until $r_k$ is $O(\sqrt{d})$.

---

> > > > > ### Author Response · Authors · 2023-08-21
> > > > > **Thank you for the question!**
> > > > >
> > > > > This is indeed a good point. Alternate selections of $k$ (e.g., $k = M/2$) and their respective generalizations appear to yield order optimality $r_k=O(\sqrt{d})$. Interestingly, this approach could reduce complexity while still achieving order optimality -- which could be studied independently in future work. However, in this work, our main concentration is on **exact optimality**. In the case of the rotated simplex codebook, we demonstrate that the $k$-closest encoding, with a carefully chosen $k$, attains this exact optimal.
> > > > >
> > > > > We thank the reviewer for asking these clarifying questions. As we approach the end of the discussion period, we hope we managed to address reviewers' concerns and questions.

---

### Official Review · Reviewer_eu7R · 2023-07-07

**Soundness:** 3 good
**Presentation:** 3 good
**Contribution:** 3 good
**Rating:** 7
**Confidence:** 3

**Summary:**

This paper studies the mean estimation problem under communication and local differential privacy constraints in the non-asymptotic (exact optimal) setting, and proposed a randomization mechanism that satisfies the identified necessary property of the exact optimality.

**Strengths:**

1. The authors proved a necessary condition that the codebook-generating distribution needs to be rotationally symmetric.

2. The authors further proposed the first exact optimality algorithm Random Rotating Simplex Coding (RRSC) that matches the necessary condition.

3. Empirical results in the paper showed that the proposed RRSC outperforms the state-of-art benchmarks (order-optimal) for the task.

4. The authors proposed interesting conjectures and clear future directions based on the design and properties of the algorithm.

**Weaknesses:**

1. line 177, "an random" to "a random"

2. The design of k-closest encoding and Theorem 3.6 (along with its proof in the Appendix) lacks the intuition on k, which seems to be valuable to explore both theoretically and empirically (as stated in the conjecture).

**Questions:**

For the choice of $k$, in the experiments are to minimize $C_k$ based on different $k$'s. How can such choice of $k$ guarantees the requirement in Theorem 3.6 (and its proof)?

**Limitations:**

Lack of intuition on k as described in weakness.

---

> ### Author Rebuttal · Authors · 2023-08-09
>
> We thank the reviewer for carefully reading our manuscript and providing positive feedback on our contributions. Specifically, we are happy to see that the reviewer recognized our proposed algorithm, Random Rotating Simplex Coding (RRSC), as the first exact-optimal scheme for distribution mean estimation problem under privacy and communication constraints; appreciated the significant empirical improvements over the existing order-optimal schemes; and found the future directions and conjectures valuable to the community. The reviewer also raised some important questions that we would like to clarify below.
> ### **The optimal choice of $k^\star$**
> We understand that it was not very clear how to select the right value for $k$ that satisfies the conditions in Theorem 3.6. We will try to elaborate on this here and will add this discussion in the final version of the manuscript as well. The optimal $k^\star$ is the minimum achiever of Equation (30) that minimizes $r_k = \frac{ke^\epsilon + M-k}{e^\epsilon-1} \sqrt{\frac{M-1}{M}}\frac{1}{C_k},$
> since the distortion is $r_k^2-1$ (as given in Eq (90) in the proof of Theorem 3.6. in Appendix E) -- i.e., the distortion is minimized if we minimize $r_k$.
>
> In practice, to address the optimization problem outlined above, an approximation of $C_k$ is necessary,  defined as the anticipated sum of the top-$k$ coordinates of a uniformly random vector $a\in\mathbb{S}^{d-1}$. For every value of $k$, we can effectively estimate $C_k$ by obtaining a sufficient number of samples of uniform random vectors $\{a_i\}$.Then, we calculate the average of the sum of the top-$k$ components for each individual vector $a_i$. Given this efficient approximation of $C_k$ for $k$, the algorithm we provide in the pdf rebuttal file (Algorithm 1) finds $k^\star$ that minimizes the $r_k$. We also provide a plot of the optimal $k^\star$ (found by Algorithm 1 in the attached pdf) as a function of $M$ for a combination of parameters in Figure 1 of the attached pdf.
>
> In the final version of the manuscript, we will add this algorithm to show the precise procedure to find the optimal $k^\star$ value. We thank the reviewer for pointing out the unclarity around the choice of $k^\star$.
> ### **Intuition on $k$**
> The intuition on the choice of $k$ can be developed from PrivUnit [1] -- the exact-optimal scheme for the same problem without the communication constraint. For a specific private vector $v\in\mathbb{S}^{d-1}$, PrivUnit assigns a higher probability to vector $w$
> that is near $v$, i.e.,  to the vector $w$ that satisfies $\langle v, w\rangle > \tau$ for some chosen threshold $\tau$. In a similar fashion, our suggested scheme, Random Rotating Simplex Coding (RRSC), assigns a higher probability to the $k$-nearest codewords. The parameter $k$ in our scheme essentially corresponds to a threshold $\tau$ in PrivUnit.
>
> In our previous response above, we have explicitly stated that we find the optimal value $k^\star$ by choosing $k$ that minimizes $r_k$,
> as described in equation (30). To identify such a $k^\star$, we must be capable of calculating $C_k$ for each $k$, a task that can be carried out simply and efficiently through Monte Carlo methods as described above and in Algorithm 1 in the attached rebuttal in pdf format. We also would like to note that estimating the optimal $k^\star$ in RRSC is significantly more straightforward and efficient
> than estimating the optimal parameters in PrivUnit.
>
> [1] A. Bhowmick, J. Duchi, J. Freudiger, G. Kapoor, and R. Rogers. Protection against reconstruction and its applications in private federated learning. arXiv preprint arXiv:1812.00984, 335 2018.
> ### **A typo on line 177**
> We sincerely thank the reviewer for the careful reading of our manuscript. We will fix the typo ("an random") on line 177 to "a random".
>
> ---
> We thank the reviewer for the careful reading of our manuscript and for pointing us to the unclear parts of the proposed scheme. We hope we have addressed the reviewer's comments, and we will add the necessary discussions in the final version of the manuscript, which we believe will improve the manuscript. We are happy to discuss any remaining questions or concerns the reviewer may have.

---

> > ### Comment · Reviewer_eu7R · 2023-08-14
> >
> > Thank you for your response and addressing my concerns, I will keep my score.

---

> > > ### Author Response · Authors · 2023-08-21
> > > **Thank you for your response to our rebuttal.**
> > >
> > > We thank the reviewer for their time reading our rebuttal.

---

### Official Review · Reviewer_FeU2 · 2023-07-07

**Soundness:** 3 good
**Presentation:** 4 excellent
**Contribution:** 3 good
**Rating:** 7
**Confidence:** 2

**Summary:**

This paper focuses on the problem of distributed mean estimation under local differential privacy (DP) and communication constraints, with shared randomness between users and the server. Previous works either achieve exact optimal mean squared error (MSE) using $O(d)$ bits or achieve order-optimal MSE with a large constant using $O(\varepsilon)$ bits. In this paper, the authors aim to tackle the same problem under shared randomness, aiming for both exact optimal MSE and communication efficiency. The proposed solution approaches the problem as a lossy compression problem. The authors demonstrate that the optimal scheme can be represented by a codebook through random coding. Additionally, they establish that the exact-optimal codebook-generating distribution must be rotationally symmetric. Empirically, the authors demonstrate that the proposed methods outperform existing approaches.

**Strengths:**

1. This paper is very technical. The paper is clearly written and lays out its contributions succinctly.

2. The proposed framework achieves exact optimality in terms of both MSE and communication efficiency.


**Weaknesses:**

It would be better if the authors could discuss why shared randomness is necessary to achieve exact optimality.

**Questions:**

See weaknesses

**Limitations:**

The limitations are discussed. This paper does not have a negative societal impact

---

> ### Author Rebuttal · Authors · 2023-08-09
>
> We thank the reviewer for their time in reading our manuscript carefully and for providing positive feedback. We are glad that the reviewer found our contributions valuable and liked the organization and writing of the manuscript. We discuss their point on shared randomness below.
>
> ### **Shared randomness**
> In our Randomly Rotating Simplex Coding (RRSC) scheme, the shared randomness is used to generate random rotation matrices that will be necessary for both encoding and decoding. We note that shared randomness is actually necessary for *any unbiased mean estimation scheme* that compresses the $d$-dimensional vectors to less than $b=\log d-2$ bits (or equivalently any unbiased mean estimation scheme that uses less than $M=2^b=O(d)$ codewords) as shown in [1, Corollary~5.1]. We will add this clarification, together with the relevant reference, in the final version of the manuscript.
>
> [1] Chen, Kairouz, and Ozgur, "Breaking the Communication-Privacy-Accuracy
> Trilemma", IEEE Transaction on Information Theory, 2023.
>
> ---
> We again thank the reviewer for the careful reading of our manuscript and for asking an important clarifying question. We will include the above discussion on shared randomness in the final manuscript. We hope that we addressed the reviewer's point sufficiently. We are happy to discuss any remaining questions the reviewer may have.

---

> > ### Comment · Reviewer_FeU2 · 2023-08-15
> >
> > Thank you for your response and addressing my questions.

---

> > > ### Author Response · Authors · 2023-08-21
> > > **Thank you for your response to our rebuttal.**
> > >
> > > We thank the reviewer for their time evaluating our manuscript and reading our rebuttal.

---

### Official Review · Reviewer_69ED · 2023-07-10

**Soundness:** 3 good
**Presentation:** 2 fair
**Contribution:** 3 good
**Rating:** 6
**Confidence:** 3

**Summary:**

This work considers the problem of distributed mean estimation and aims to obtain "exact optimal" estimators under communication, (local) differential privacy and utility constraints. Exact optimality here means that instead of focusing on the order of complexity, the focus is also on the constants as well. Prior work either focused on "order optimality" or were exactly optimal for privacy and utility but only order optimal [Feldman-Talwar'11], [Shah-Chen-Balle-Kairouz-Theis'22]. This work achieves exact optimality through a k-closest encoding and a randomly generated codebook shared between the server and users.



**Strengths:**

They obtain exact optimality for privacy-utility-communication tradeoffs of mean estimation under l_2 norm error. This improves upon prior work that could only obtain order optimality or exact optimality but only for some parameters.

Their work unifies the framework that existed in prior work [Feldman-Talwar'11], [Shah-Chen-Balle-Kairouz-Theis'22].

Their experiments show significant improvement in communication budget required compared to the previous work, especially in the setting where the number of bits is small.

The theorem regarding optimality of rotationally symmetric shared random codebooks could be of independent interest.

**Weaknesses:**

There is no "main theorem" that sums up the results in the main results section for mean estimation, and I think it's necessary to include such a theorem.

I think exact optimality compared to order optimality is a more niche setting. That being said, that's not necessarily a weakness, and it could be impactful in practice.

**Questions:**

I think part of the paragraph **unified framework** in the discussion section could be mentioned in the related work section as well.

As mentioned in the previous section a main theorem that clearly states the trade offs this work obtain for mean estimation should be included. The score currently given is assuming that such a theorem will be included.

**Limitations:**

The authors have adequately addressed the limitations of their work.

---

> ### Author Rebuttal · Authors · 2023-08-09
>
> We thank the reviewer for carefully reading our manuscript and for providing constructive feedback to further improve it. Specifically, we are glad that the reviewer found the theoretical and empirical improvements over the existing order-optimal schemes impressive; and highlighted the significance of the rotational symmetry condition for the optimal codebook. Upon reviewer's suggestions, we will add the following theorems in the final manuscripts, which we think will improve the clarity of the results. Therefore, we thank the reviewer for these valuable suggestions.
> ### **A main theorem that sums up the main results**
> In the theorem below, we put together the main results in Section 3 that summarize the main theoretical contributions of our work. We will provide this theorem in the final manuscript.
>
> **Theorem 1:** *There exists a canonical protocol with a rotationally symmetric random codebook
> that achieves the exact-optimal worst-case error among all unbiased $\ell_2$-mean estimation schemes
> under $\varepsilon$-local differential privacy (LDP) and $b$-bit constraint simultenously. Moreover, there exists a $k$ such that the $k$-closest encoding algorithm in Section 3.1 in the manuscript is the exact-optimum unbiased scheme for a rotationally symmetric simplex codebook under $\varepsilon$-LDP and $b$-bit constraint. The optimal $k$ that achieves this exact optimality is found by minimizing $r_k$ in Eq. (30) since the error is $r_k^2-1$ (as shown in Eq. (90) in Appendix E) --- see Algorithm 1 in the attached pdf rebuttal for the precise description on how to find the optimal $k$.*
>
> ### **A theorem that clearly states the communication-privacy-utility trade-offs**
> In this work, we show compelling evidence that our proposed scheme, Randomly Rotating Simplex Coding (RRSC), achieves exact optimality. Upon the reviewer's suggestion, we also provide the theorem and its proof below that shows the order optimality (which is actually a weaker statement than the exact optimality result we show in the manuscript) of RRSC with a clear statement of the error.
>
> **Theorem 2:** *In the region of interest where $1 \leq \varepsilon \leq b \leq d$, RRSC($k^\star$), i.e., RRSC with the optimal choice of k, achieves an error of $\frac{1}{n} \left ( (k^\star)^2   (\frac{e^{\varepsilon} + 1 / q -1}{e^{\varepsilon} -1} )^2 \frac{M-1}{M} \frac{1}{C_{k^\star}^2} - 1 \right)$, where $q=k^\star/M<1$. This corresponds to an error that scales with $\frac{d}{n}$ -- satisfying the order optimality.*
>
> The rough proof is the following. As stated in Eq. (60) in Appendix E, the $\ell_2$ error is $r_k^2-1$, where $r_k$ is precisely defined in Eq. (30). The rest of the proof follows from the fact that (1) the optimal $k^\star$ which is found by minimizing $r_k$ (hence the error) satisfies $k^\star = O(M)$ and (2) $C_{k^\star}=O(k^\star/\sqrt{d})$. The algorithm to find the optimal $k$ is given in Algorithm 1 in the attached pdf, where we also provide a figure that justifies $k^\star = O(M)$. We will include this theorem and its full proof in the final manuscript.
> ### **The motivation to study exact optimality**
> As the reviewer mentioned, there are already existing schemes that achieve order optimality, but it was unclear how much these schemes could be improved in the non-asymptotic regime and how far to the optimal the pre-constant of the estimation error is. It is worth noting that in many practical scenarios (such as federated learning and analytics), the constant factor of the estimation error does significantly affect the end-to-end performance. As a result, in this work, our goal is to bridge this gap by (1) specifying exact-optimality conditions, (2) proving the exact optimality of a novel scheme, and (3) demonstrating that there is indeed a non-trivial gap between this exact-optimal scheme and the previous order-optimal approaches. Therefore, we hope to convey to the community the fundamental limits of the distributed mean estimation problem under both privacy and communication constraints and provide a strong baseline for further studies. We are glad that the reviewer also finds this perspective valuable.
> ### **Moving the unified framework discussion to the related work section**
> We thank the reviewer for this suggestion. We agree that this discussion suits well to the related work section as well. In the final version of the manuscript, we will provide a high-level idea of how we unify the existing schemes in the related work section and give further details in the discussion after we cover the main results.
>
> ---
> We again thank the reviewer for the careful reading of our manuscript and for providing valuable suggestions to improve it further. We will include the additional theorems in the final manuscript and will carry the part of the discussion on unifying the framework to the related work section as suggested by the reviewer. We hope that we addressed the reviewer's comments sufficiently. We are happy to discuss any remaining questions or concerns the reviewer may have.

---

> > ### Comment · Reviewer_69ED · 2023-08-20
> >
> > I thank the authors for their response. I will keep my score.
> >
> > For the final version of the paper, I also suggest including a discussion paragraph that compares the exact optimal error rate provided here with the guarantees of the previous work about order optimal schemes. The authors mention that in practice the constant factor difference makes a difference, and have demonstrated this through experiments. However, it would be interesting to see how much they differ theoretically.

---

> > > ### Author Response · Authors · 2023-08-21
> > > **Thank you for your response to our rebuttal.**
> > >
> > > We thank the reviewer for the suggestion. We will include the suggested paragraph in the final manuscript.

---

### Author Rebuttal · Authors · 2023-08-09

We would like to thank all the reviewers for carefully reading our paper, and providing positive and constructive feedback to further improve it. All the reviewers seem to appreciate the technical contributions of the paper in studying the exact optimality of the distributed mean estimation problem under privacy and communication constraints. Specifically, all the reviewers found theoretical and empirical improvements over the existing order-optimal schemes worth praising. Moreover, Reviewer 69ED found one of our main results, namely, the exact optimality of the rotationally symmetric codebook, worthy of independent interest;  Reviewer FeU2 liked the clear and fluent presentation of the results and the organization of the manuscript; Reviewer eu7R highlighted that the proposed algorithm RRSC outperforms the state-of-the-art schemes and stressed the importance of the conjectures and future directions in the manuscript to the community; Reviewer jPit found the k-closest encoding algorithm, which we prove to be the exact-optimal approach for a simplex codebook, impressive, clean, and worthy of independent interest in general; and lastly Reviewer JsmV underlined the fact that previous order-optimal solutions can be viewed as special cases of the general setup we propose in this work -- namely the canonical protocol with rotationally symmetric codebook.

The reviewers also shared valuable suggestions that we believed improved the overall quality of our manuscript. We explain how we address them and how we will revise the manuscript in our separate responses to each reviewer below. We also submit another rebuttal file in pdf format to share an additional algorithm and a figure as part of our response to some comments.

---

### Decision · Program_Chairs · 2023-09-21

**Decision:**

Accept (poster)

**Comment:**

The paper studies distributed mean estimation in the local differential privacy model. The authors propose a near optimal method in terms of privacy-utility-communication tradeoff. While the previous work had established asymptotically tight bounds, the authors have made progress in achieving the optimal constants for this fundamental problem.

The reviewers were all positive about this paper. The authors have adequately addressed the technical questions raised by the reviewers.